# Fetal growth delay caused by loss of non-canonical imprinting is resolved late in pregnancy and culminates in offspring overgrowth

Ruby Oberin[1†], Sigrid Petautschnig[1†], Ellen G Jarred[1], Zhipeng Qu[2], Tesha Tsai[1‡], Neil A Youngson[1,3], Gabrielle Pulsoni[1], Thi T Truong[4], Dilini Fernando[1], Heidi Bildsoe[1], Rheannon O Blücher[1], Maarten van den Buuse[5], David K Gardner[4], Natalie A Sims[6], David L Adelson[2], Patrick S Western[1*]

[1]Centre for Reproductive Health, Hudson Institute of Medical Research and Department of Molecular and Translational Science, Monash University, Clayton, Australia; [2]Department of Molecular and Biomedical Sciences, School of Biological Sciences, University of Adelaide, Adelaide, Australia; [3]School of Biomedical Sciences, University of New South Wales, Sydney, Australia; [4]School of BioSciences, University of Melbourne, Parkville, Australia; [5]School of Psychology and Public Health, La Trobe University, Melbourne, Australia; [6]Bone Cell Biology and Disease Unit, St. Vincent's Institute of Medical Research and Department of Medicine at St. Vincent's Hospital, University of Melbourne, Fitzroy, Australia

*For correspondence:
patrick.western@hudson.org.au

[†]These authors contributed equally to this work

Present address: [‡]School of Health and Biomedical Sciences, RMIT University, Bundoora, Australia

Competing interest: The authors declare that no competing interests exist.

**Abstract** Germline epigenetic programming, including genomic imprinting, substantially influences offspring development. Polycomb Repressive Complex 2 (PRC2) plays an important role in Histone 3 Lysine 27 trimethylation (H3K27me3)-dependent imprinting, loss of which leads to growth and developmental changes in mouse offspring. In this study, we show that offspring from mouse oocytes lacking the PRC2 protein Embryonic Ectoderm Development (EED) were initially developmentally delayed, characterised by low blastocyst cell counts and substantial growth delay in mid-gestation embryos. This initial developmental delay was resolved as offspring underwent accelerated fetal development and growth in late gestation resulting in offspring that were similar stage and weight to controls at birth. The accelerated development and growth in offspring from *Eed*-null oocytes was associated with remodelling of the placenta, which involved an increase in fetal and maternal tissue size, conspicuous expansion of the glycogen-enriched cell population, and delayed parturition. Despite placental remodelling and accelerated offspring fetal growth and development, placental efficiency, and fetal blood glucose levels were low, and the fetal blood metabolome was unchanged. Moreover, while expression of the H3K27me3-imprinted gene and amino acid transporter *Slc38a4* was increased, fetal blood levels of individual amino acids were similar to controls, indicating that placental amino acid transport was not enhanced. Genome-wide analyses identified extensive transcriptional dysregulation and DNA methylation changes in affected placentas, including a range of imprinted and non-imprinted genes. Together, while deletion of *Eed* in growing oocytes resulted in fetal growth and developmental delay and placental hyperplasia, our data indicate a remarkable capacity for offspring fetal growth to be normalised despite inefficient placental function and the loss of H3K27me3-dependent genomic imprinting.

## Editor's evaluation

This important study shows that a lack of Polycomb-dependent epigenetic programming in the oocyte and early embryo influences the developmental trajectory through gestation in the mouse. The authors provide convincing evidence for a two-phase outcome of early growth restriction followed by enhancement, addressing previous inconsistencies in the field. The work establishes a link between the function of a protein in oocytes and programming of fetal growth and placental function in late gestation, though the underlying functional relationshsip between increased fetal growth and placental function will require further investigation. This manuscript will interest scientists within the fields of developmental biology and epigenetics.

## Introduction

Epigenetic mechanisms orchestrate tissue development, body patterning, and growth by regulating chromatin accessibility and expression of developmental genes in embryonic and extraembryonic tissues (*Maccani and Marsit, 2009*; *Sun et al., 2021*). H3K27me3 is a repressive epigenetic modification catalysed by PRC2, which is composed of the core components, EED, EZH1/2, SUZ12, and RBBP4/7, all of which are essential for histone methyltransferase activity (*Pasini et al., 2004*; *Faust et al., 1995*; *O'Carroll et al., 2001*; *Shen et al., 2008*; *Glancy et al., 2021*). Global deletion of any of the individual genes encoding these PRC2 subunits results in embryonic lethality in mice, while conditional deletions of *Eed, Ezh2,* or *Suz12* result in altered patterning and organ function in a range of tissues (*Pasini et al., 2004*; *Prokopuk et al., 2018*; *Boyer et al., 2006*; *Hemming et al., 2014*; *Dudakovic et al., 2015*; *Majewski et al., 2010*; *Liu et al., 2019*; *Sun et al., 2018*; *Miao et al., 2020*). In humans, de novo germline mutations in *EED, EZH2,* or *SUZ12* result in Cohen Gibson, Weaver, and Imagawa-Matsumoto syndromes, which all involve overgrowth, skeletal dysmorphologies, and cognitive abnormalities (*Cohen and Gibson, 2016*; *Tatton-Brown et al., 2011*; *Cohen et al., 2016*; *Cooney et al., 2017*; *Imagawa et al., 2018*; *Imagawa et al., 2017*; *Tatton-Brown et al., 2013*).

In mouse oocytes, PRC2 is required for non-canonical H3K27me3-imprinting, whereby H3K27me3 silences the maternal allele of several genes resulting in paternal allele-specific expression of the affected genes in pre-implantation embryos and extraembryonic tissues (*Hanna and Kelsey, 2021*; *Inoue et al., 2017*). Oocyte-specific deletion of *Eed* results in loss of H3K27me3-imprinting in preimplantation embryos and extraembryonic ectoderm causing growth restriction and male-biased lethality in mid-gestation offspring (*Inoue et al., 2018*). In contrast to growth restriction, another study found that postnatal offspring were overgrown following deletion of *Eed* in oocytes (*Prokopuk et al., 2018*). Loss of H3K27me3-imprinting in somatic cell nuclear transfer (SCNT) embryos results in placental hyperplasia associated with bi-allelic expression of H3K27me3 imprinted genes, including *Xist, Slc38a4,* a micro-RNA cluster within *Sfmbt2* (*C2MC*), *Gab1,* and *Smoc1* (*Matoba et al., 2018*; *Inoue et al., 2020*; *Wang et al., 2020*; *Xie et al., 2022*; *Matoba et al., 2022*). Moreover, deletion of *Eed* in oocytes caused placental hyperplasia and placental hyperplasia was partially or completely rescued by homozygous deletion of *Slc38a4* or *Sfmbt2* together with *Eed* in oocytes. In addition, homozygous maternal deletion of *Xist* and *Eed* rescued male-biased lethality and partially rescued developmental delay and survival in offspring (*Matoba et al., 2022*).

Here, we show that loss of EED in the oocyte results in initial fetal growth restriction followed by fetal growth recovery and perinatal offspring overgrowth. While fetuses derived from *Eed*-null oocytes were initially developmentally delayed and growth restricted, they underwent accelerated growth and development facilitating the birth of pups at normal weight from pregnancies of normal gestational length. Moreover, remedial fetal growth occurred in the presence of placental hyperplasia, reduced placental efficiency, low fetal blood glucose, and normal fetal blood amino acid and metabolite levels. Together, this work reveals a complicated offspring growth trajectory that involved in utero resolution of fetal growth restriction and developmental delay, followed by perinatal overgrowth, despite the loss of H3K27me3-dependent imprinting caused by *Eed* deletion in oocytes.

## Results

### Loss of maternal EED compromised mid-gestation survival in offspring

Previous studies demonstrated that deletion of *Eed* in oocytes resulted in contradictory findings of early embryonic developmental delay (*Inoue et al., 2018*) and early postnatal offspring overgrowth (*Prokopuk et al., 2018*). To understand how these outcomes are realised, C57BL/6 *Eed*^fl/fl^ or *Eed*^fl/wt^ females were mated to males that were transgenic for *Zp3Cre* to generate females producing *Eed* wild-type (wt), *Eed* heterozygous (het), or *Eed*-homozygous null (hom) oocytes (*Prokopuk et al., 2018*). Mating of these females to isogenic C57BL/6 *wt* males allowed us to generate isogenic wild-type or heterozygous offspring as previously described (*Figure 1A*; *Prokopuk et al., 2018*). Heterozygous offspring generated from *Eed* heterozygous oocytes (HET-het offspring) or from *Eed* homozygous null oocytes (HET-hom offspring) are isogenic. However, the HET-hom offspring were generated from oocytes that completely lacked functional EED and the HET-het offspring were generated from oocytes that had one functional copy of EED and maintained essentially normal gene repression (*Prokopuk et al., 2018*; *Jarred et al., 2022*). As they are isogenic and heterozygous for *Eed*, comparison of HET-hom with HET-het offspring allowed the identification of differences that resulted specifically from a loss of EED in oocytes in the absence of confounding genetic differences (*Figure 1A*). In addition, we generated wild-type offspring from *Eed-wt* and *Eed-het* oocytes to compare *WT-wt* and *WT-het* offspring with HET-het and HET-hom offspring, providing controls for differences generated by *Eed* heterozygosity in offspring (*Figure 1A*).

We used automated time-lapse imaging of individual embryos derived from *Eed*-wt and *Eed*-hom oocytes to track development from zygote to hatched blastocyst stages. Embryos from *Eed*-hom oocytes reached the two cell stage 1.08 hr earlier than embryos from *Eed*-wt oocytes (*Figure 1B*), but blastocyst expansion and hatching took 2.67 hr longer in HET-hom compared to WT-wt embryos (*Figure 1C*). All other developmental milestones to blastocyst stage were similar between genotypes (*Figure 1B*). Although, viability to blastocyst stage was lower in embryos from females producing *Eed*-hom compared to *Eed*-wt oocytes (72.17% vs 88.67%, *Figure 1D*), the proportion of expanded and hatched blastocyst stage embryos did not differ significantly (*Figure 1—figure supplement 1A*).

To determine whether the cell content of expanded blastocysts was affected, we performed cell counts in whole blastocysts at the end of the culture period using differential staining of the inner cell mass (ICM) and trophectoderm (TE) cells (*Gardner and Lane, 2014*; *Figure 1—figure supplement 1B*). This revealed that the overall cell content of embryos from *Eed*-hom females was lower than in *Eed*-wt controls (*Figure 1E*). The number of TE cells was not significantly altered, but the ICM contained fewer cells in HET-hom embryos compared to WT-wt controls (*Figure 1F*; *Figure 1—figure supplement 1C*), indicating that the ICM was smaller in embryos from oocytes that lacked EED compared to oocytes that maintained EED function (*Figure 1F*). As previous reports indicated that maternal deletion of *Eed* did not compromise development to the blastocyst stage, or cause apoptosis in blastocysts (*Inoue et al., 2018*), it is likely that lower proliferation explains the reduction we observed in blastocyst cell number, which affected the ICM rather than the TE.

To assess the effect of oocyte-specific *Eed* deletion on fetal survival and throughout pregnancy, we compared the number of live fetuses and litter size in offspring generated from *Eed-hom*, *Eed*-het, and *Eed*-wt oocytes at multiple stages during gestation and after birth. Using *Zp3*Cre to delete *Eed* in oocytes, we found no difference between genotypes in the number of implantations of live embryos in four pregnancies from *Eed*-null oocytes analysed at embryonic day (E)9.5 (*Figure 1G*), indicating that HET-hom preimplantation embryos implanted at similar rates and progressed through gastrulation. However, consistent with previous findings (*Glancy et al., 2021*; *Hanna and Kelsey, 2021*), the number of live fetuses at E12.5, E14.5, E17.5, and the number of live born pups was significantly lower for *Eed*-hom mothers than for *Eed*-het and *Eed*-wt control females (*Figure 1G*). In addition, 71.43% of live fetuses in E17.5 *Eed*-hom pregnancies were female, indicating male-biased lethality of HET-hom offspring (*Figure 1H*), consistent with previous studies (*Inoue et al., 2018*; *Harris et al., 2019*).

### Loss of EED in oocytes results in developmental delay followed by fetal growth recovery and postnatal overgrowth

While a previous study observed that deletion of *Eed* in oocytes caused embryo loss by E6.5 (*Matoba et al., 2018*), we observed loss of HET-hom offspring between E9.5 and E12.5, indicating that embryo survival and/or placental function was compromised. We, therefore, assessed growth

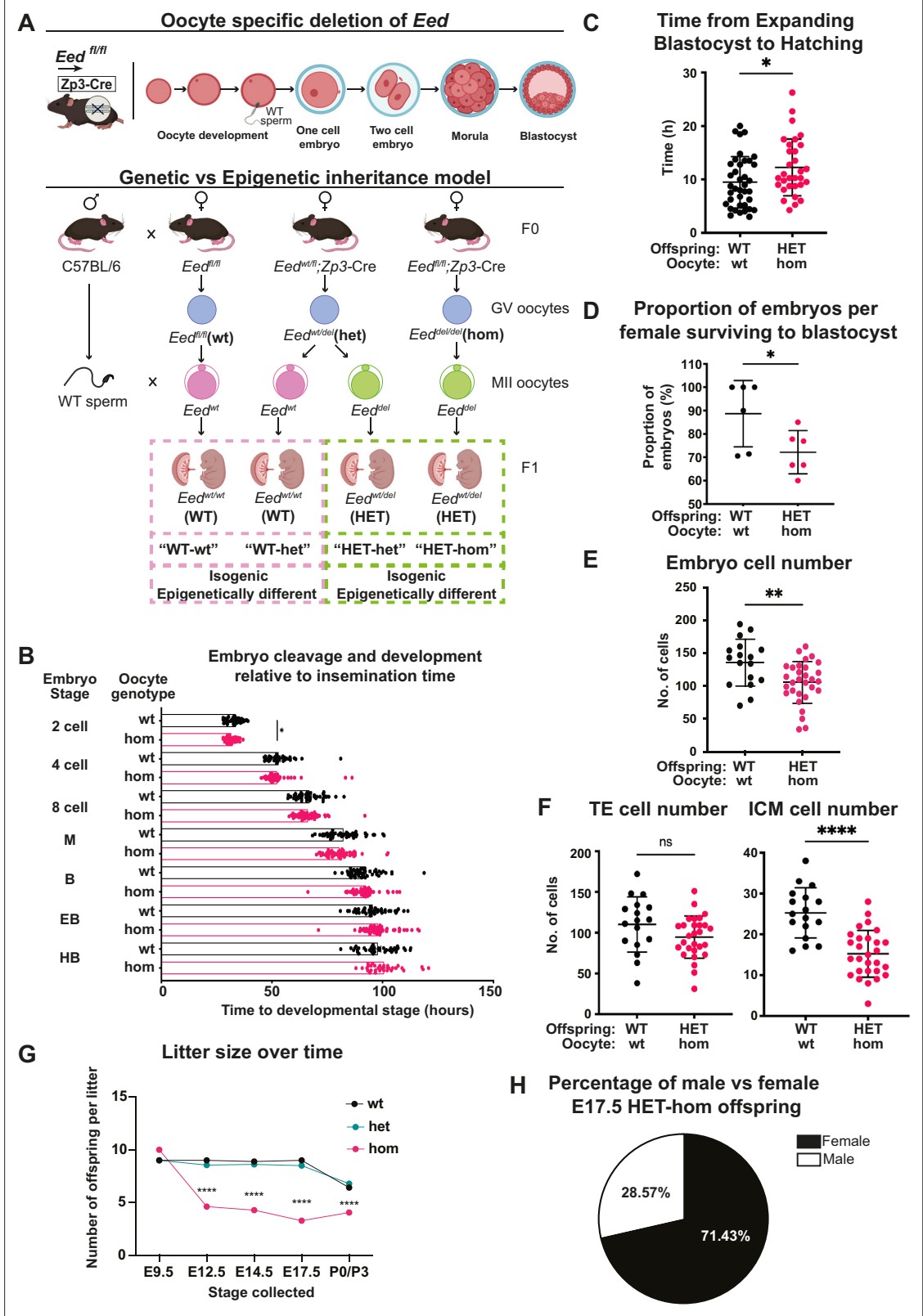

**Figure 1.** Maternal *Embryonic ectoderm development* (*Eed)* deletion impacted pre-implantation development. (**A**) Model used to generate heterozygous isogenic offspring from oocytes that lacked or retained *Eed*. Females producing *Eed* wild-type (*Eed*-wt), *Eed* heterozygous (*Eed*-het), and *Eed*-homozygous (*Eed*-hom) oocytes were mated with wild-type males to produce offspring from oocytes with wt or het EED-dependent programming or from oocytes that lack EED-dependent programming, respectively. *Eed*-wt oocytes produce wild-type (WT) offspring (WT-wt control), *Eed*-het oocytes

*Figure 1 continued on next page*

*Figure 1 continued*

produce WT or HET offspring (WT-het and HET-het) from GV oocytes, and *Eed*-hom oocytes produce HET offspring (HET-hom). HET-het and HET-hom offspring are isogenic but were derived from oocytes that had different EED-dependent programming. (**B**) Cell cleavage and development times of embryos from *Eed*-wt or *Eed*-hom oocytes in ex vivo culture. Data is presented as time to reach 2-, 4-, 8-cell, morula (M), blastocyst (B), expanded blastocyst (EB), and hatched blastocyst (HB). *p<0.05, two-tailed student's t-test, n=49 embryos from *Eed*-wt oocytes, 57 embryos from *Eed*-hom oocytes. (**C**) Time taken for expanded blastocysts to hatch. Data represents the time difference between hatched blastocysts and expanded blastocysts. *p<0.05, n=38 embryos from *Eed*-wt oocytes, 31 embryos from *Eed*-hom oocytes. (**D**) Proportion of embryos from *Eed*-wt or *Eed*-hom oocytes surviving to blastocyst. *p<0.05, Data represent the proportion of surviving embryos from each female, n=6 females. (**E**) Total number of cells per embryo. **p<0.005, n=17 embryos from *Eed*-wt oocytes, 30 embryos from *Eed*-hom oocytes. (**F**) Proportion of trophectoderm (TE) and inner cell mass (ICM) cells per embryo. Data represents the mean proportion of cells allocated to TE versus ICM for embryos from *Eed*-wt or *Eed*-hom oocytes. **p<0.05, n=17 embryos from *Eed*-wt oocytes, 28 embryos from *Eed*-hom oocytes. (**G**) Offspring litter size over time. ****p<0.0001. (**H**) Pie chart depicting the proportion of male and female HET-hom offspring at E17.5. N=32. For (**B–G**) a two-tailed student's t-test was used. Error bars: mean ± SD.

The online version of this article includes the following figure supplement(s) for figure 1:

**Figure supplement 1.** Maternal *Embryonic ectoderm development* (*Eed*) deletion impacted pre-implantation development.

and development of surviving mid-late gestation offspring. At E9.5, HET-hom offspring were small compared to WT-wt offspring (*Figure 2A*) and combined offspring and extraembryonic tissue weights were significantly lower than HET-het, WT-het, and WT-wt controls (*Figure 2—figure supplement 1A*). At E12.5 HET-hom embryos contained fewer tail somites and had delayed inter-digital tissue regression in foot plates at E14.5 compared to HET-het, WT-het, and WT-wt controls (*Figure 2B*, *Figure 2—figure supplement 1B*), demonstrating that HET-hom growth and development were retarded during early-mid gestation development.

As in utero developmental delay contrasted markedly with our previous observation that P2 HET-hom offspring were overgrown (*Prokopuk et al., 2018*) we examined the temporal trajectory of HET-hom offspring by measuring fetal weight at E12.5, E14.5, E17.5, and E18.5 and pup weight on the day of birth (P0) and at P3. HET-hom offspring were significantly lighter than HET-het, WT-het, and WT-wt controls at E12.5, E14.5, E17.5, and E18.5, but by P0 and P3 HET-hom offspring were heavier than isogenic HET-het controls (*Figure 2C–H*). However, by monitoring plug date and delivery day, we also observed that gestational length was extended in 60% of pregnancies from *Eed*-hom oocytes. Of 23 litters, 9 (40%) were born on E19.5, 7 (30%) were born on E20.5, and 7 (30%) were born on E21.5. While all pups born on E19.5 survived, 30% and 67% of offspring from pregnancies extended by one or two days respectively, were found dead. In contrast, although one litter from an *Eed*-het pregnancy was delivered on E20.5, 23 (9 wt and 14 het) were delivered on E19.5 (*Figure 2I*).

To determine if the extended gestational length of pregnancies from *Eed*-hom oocytes affected HET-hom offspring weight, we examined pup weight in pregnancies delivered on E19.5 only. This revealed that HET-hom offspring delivered on E19.5 were similar in weight and developmental stage to controls (*Figure 2J*). Moreover, comparison of data for all surviving HET-hom pups delivered on E19.5-E21.5 (*Figure 2G*) with those only born on E19.5 (*Figure 2J*) revealed that increased gestational length contributed to the overall increased pup weight of the population at P3 (*Figure 2H*). However, ratios of HET-hom fetal weight over WT-wt, WT-het, or HET-het fetal weight at E12.5, E14.5, E17.5, E18.5, and E19.5 (excluding litters of extended gestational length) revealed that the weight deficit in HET-hom was resolved between E14.5 and birth on E19.5 (*Figure 2L*). Moreover, with the exception of the modest weight deficit, E18.5 HET-hom offspring were indistinguishable from controls at the gross morphological level. Together, these data demonstrate that the gross morphological delay and the fetal weight deficit in HET-hom offspring was resolved by E19.5 and that this occurred independently of litter size. In addition, by P3 HET-hom offspring were heavier than controls. As the overgrown HET-hom offspring in the P3 weight cohort (*Figure 2H*) included litters born at E19.5 E20.5 and E21.5, both postnatal offspring growth and extended gestation presumably contributed to the overgrowth phenotype.

We also observed increased fetal resorptions and fetal death (*Figure 2—figure supplement 1C*), raising the possibility that the reduced number of fetuses/pregnancy might result in greater maternal support per fetus leading to normalisation of pup weight at E19.5 (*Prokopuk et al., 2018*). However, there was no difference in pup weight from *Eed*-hom and control litters that contained 5–7 pups and were born on E19.5 (*Figure 2K*), indicating that resolution of fetal growth restriction in utero was at least partly independent of litter size, a finding consistent with our earlier study (*Prokopuk et al.,*

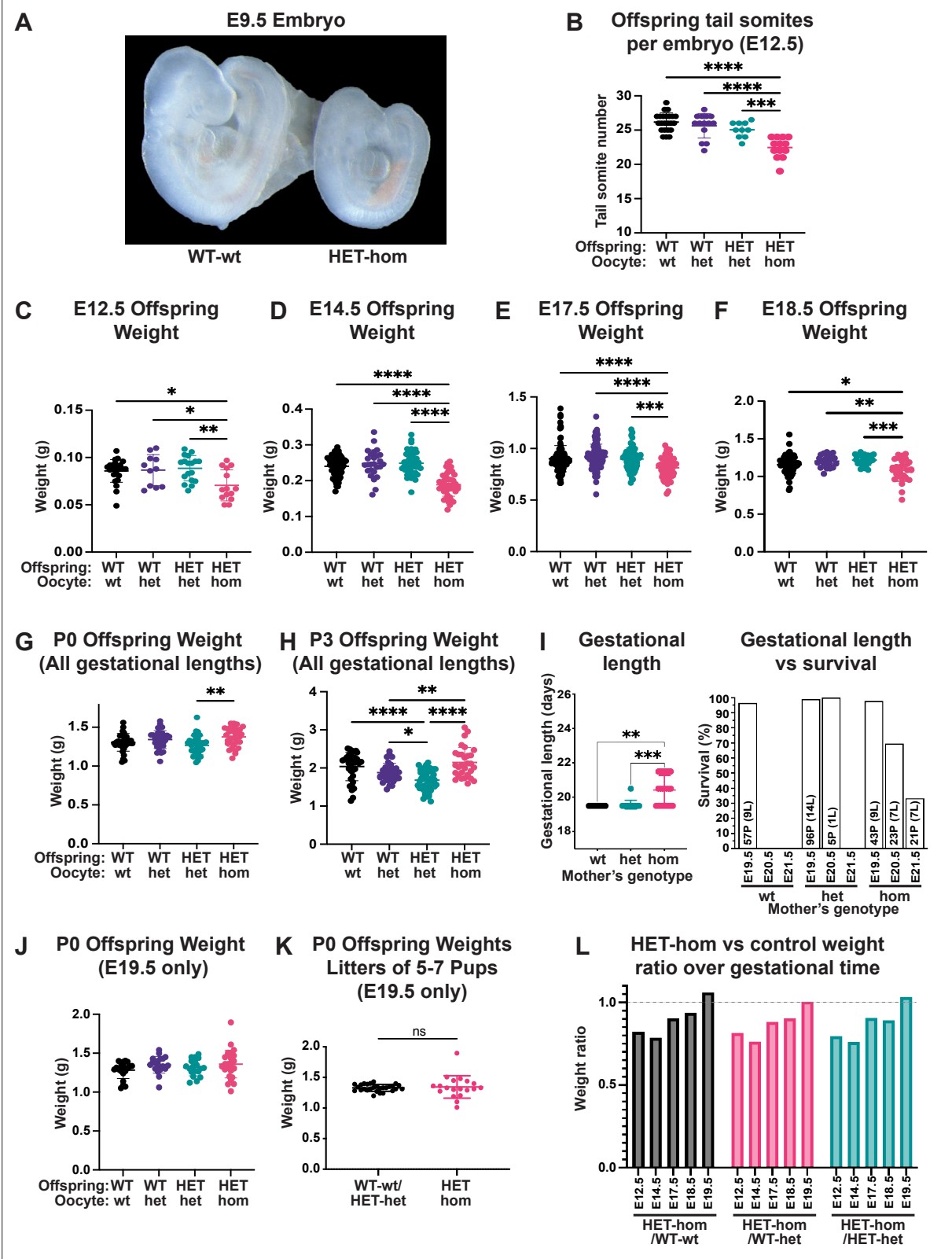

**Figure 2.** Loss of Embryonic ectoderm development (EED)-dependent oocyte programming resulted in fetal loss and offspring developmental delay. (**A**) Representative wholemount images of E9.5 embryos. Images representative of 2–3 litters/genotype. (**B**) Number of tail somites in E12.5 *Eed* WT-wt, WT-het, HET-het and HET-hom embryos. \*\*\*p<0.0005, \*\*\*\*p<0.0001, n=10–23/genotype. (**C–F**) Fetal weight at E12.5, E14.5, E17.5, and E18.5. \*\*p<0.005, \*\*\*p<0.0005, \*\*\*\*p<0.0001, n=11–117/genotype. (**G–H**) Offspring body weights at P0 and P3 from litters of all gestational lengths. \*p<0.05, \*\*p<0.005,

*Figure 2 continued on next page*

*Figure 2 continued*

****p<0.0001, n=30–50/genotype. (**I**) Gestational length and survival in litters from *Eed*-wt, *Eed*-het, and *Eed*-hom oocytes. **p<0.01, ***p<0.001. Data for each genotype include: *Eed*-wt 57 pups (P) from 9 litters (L) all born on E19.5; *Eed*-het 96 pups from 14 litters born on E19.5 and five pups from 1 litter born on E20.5; *Eed*-hom 43 pups from 9 litters born on E19.5, 23 pups from 7 litters born on E20.5, and 21 pups from 7 litters born on E21.5. (**J**) Offspring body weight for litters born on E19.5. n=19–28/genotype. (**K**) Offspring body weights of litters containing 5–7 pups and were born on E19.5. ns = non-significant difference, unpaired t-test, n=19–28/group. (**L**) Ratios of HET-hom offspring weight to WT-wt, WT-het, and HET-het control weight at E12.5, E14.5, E17.5, E18.5, and E19.5. E19.5 data included only litters born on E19.5. (**B–J**) Statistically analysed using one-way ANOVA plus Tukey's multiple comparisons. Error bars: mean ± SD.

The online version of this article includes the following figure supplement(s) for figure 2:

**Figure supplement 1.** Loss of Embryonic ectoderm development (EED)-dependent oocyte programming altered fetal and neonatal development.

*2018*). Combined, while HET-hom offspring were initially developmentally delayed and under-weight compared to their isogenic heterozygous counterparts and wild-type controls, HET-hom offspring weight was normalised by birth, and pups were overgrown by P3 (*Figure 2J and K*), an outcome facilitated by fetal growth recovery and extended gestation.

## Oocyte-specific loss of EED results in placental hyperplasia and reduced placental efficiency

To understand whether the normalisation of HET-hom fetal weight might be related to changes in placental development, we collected and weighed E12.5, E14.5, E17.5, and E18.5 placentas from HET-hom, HET-het, WT-het, and WT-wt offspring. At E12.5 placental weights were consistent across all genotypes, but at E14.5 HET-hom placentas were slightly heavier than HET-het controls (*Figure 3A–B*). By E17.5 HET-hom placentas were 58% heavier than HET-het controls (*Figure 3C*), and at E18.5 HET-hom placentas were 24% heavier than HET-het placentas (*Figure 3D*). These changes were reflected in differing HET-hom and HET-het placental growth rates over time with HET-hom placenta growth rates 0.79 and 1.81 times greater than HET-het placentas between E12.5 and E14.5 and E14.5 and E17.5, respectively (*Figure 3E–F*). Comparison of HET-hom placental to fetal weights revealed that placental growth preceded late-gestational fetal growth (*Figure 3E–F*). The HET-hom fetal growth rate was slightly lower than HET-het offspring between E12.5 and E14.5, similar between E14.5 and E18.5, but was greater between E18.5 and E19.5 so that HET-hom pups were of equivalent weight to controls at birth on E19.5 (excluding litters with extended gestation; *Figures 2C–F, J, 3E and G*). To further understand the relationship between placental function and offspring growth, we calculated the offspring body to placental weight ratio, which is indicative of placental efficiency (*Krombeen et al., 2019*). By this measure, placental efficiency was reduced at E12.5, E14.5, E17.5, and E18.5 in HET-hom offspring compared to all other genotypes (*Figure 3H–K*).

## Loss of EED in the oocyte results in altered developmental patterning of the placenta

While we observed male-biased fetal loss and that E17.5 HET-hom placentas substantially differed from controls, our data revealed that male and female fetuses and placentas were of similar weights at E14.5 and E17.5 (*Figure 3—figure supplement 1A–D*). To examine male and female placental development in more detail we stained sections from the midline of the placenta with periodic shiff (PAS; *Figure 3L*) and hematoxylin and eosin (H&E; *Figure 4A–B*) and used QuPath to quantify placental, junctional zone, decidua areas, and glycogen-enriched cells. To ensure comparative analyses between individuals, we examined sections from the middle of the placentas from E14.5 and E17.5 male and female WT-wt, WT-het, HET-het, and HET-hom offspring. Comparison of male and female samples revealed no sex-specific differences in placental weight or cross-sectional areas of placenta, decidua, or junctional zone, but the cross-sectional areas of placenta, junctional zone, and decidua were increased in HET-hom placentas (*Figure 3—figure supplement 1C–G*). Interestingly, this lack of sex-specific differences was consistent with similar fetal and placental weight between males and females (*Figure 3—figure supplement 1A–D*), indicating that male-biased lethality (*Figure 1H*) was independent of obvious sex-specific changes in the placenta or fetal growth rate.

Given the lack of sex-specific changes in placental weight or cross-sectional areas (*Figure 3—figure supplement 1*), we analysed male and female samples together. H&E and PAS staining

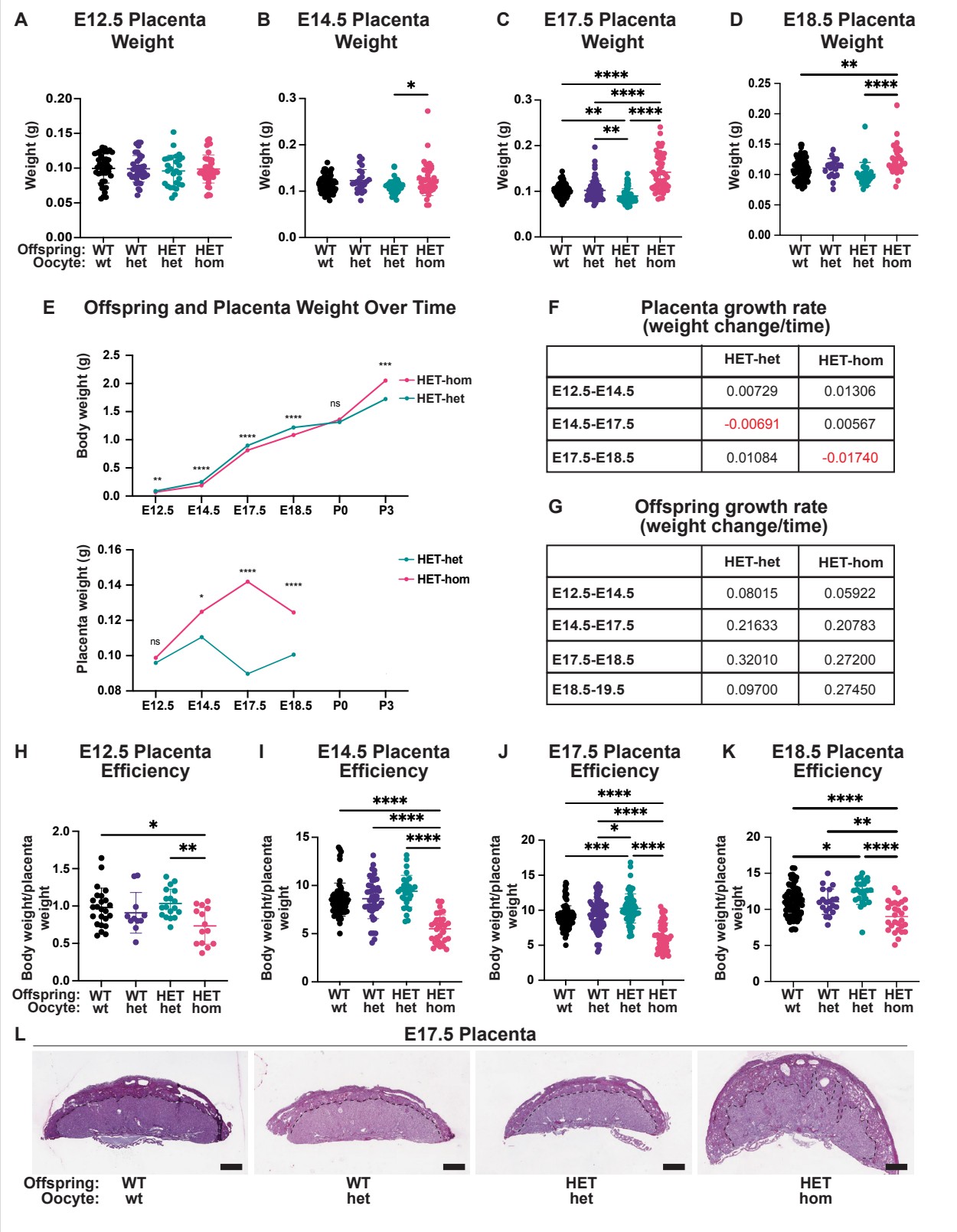

**Figure 3.** Placental differences observed in heterozygous offspring from oocytes that lacked *Embryonic ectoderm development* (*Eed*). (A–D) Placental weight at E12.5, E14.5, E17.5, and E18.5, *p<0.05, **p<0.005, ****p<0.0001, n=20–110/genotype. (E) Heterozygous offspring and placenta average weight over time. ns = non-significant difference, *p<0.05, **p<0.005, ***p<0.0005, ****p<0.0001 unpaired t-test, n=14–64/genotype. (F–G) Tables depicting *Eed* HET-het and HET-hom offspring and placenta growth rates across three to four time points throughout gestation. (H–K) Ratio of placental

*Figure 3 continued on next page*

*Figure 3 continued*

weight to fetal weight at E12.5, E14.5, E17.5, and E18.5. *p<0.05, **p<0.005, ***p<0.0005, ****p<0.0001, n=11–108/genotype. (**L**) Placental cross-sections at E17.5 stained with periodic shiff (PAS). Scale bars = 800 µm. (**A–D, H–K**). One-way ANOVA with Tukey's multiple comparisons. Error bars: mean ± SD.

The online version of this article includes the following figure supplement(s) for figure 3:

**Figure supplement 1.** Deletion of *Embryonic ectoderm development* (*Eed*) in the oocyte resulted in similar changes in male and female offspring and placental growth at E14.5 and E17.5.

indicated that the junctional zone was expanded in HET-hom placentas, with abnormal projections of PAS-stained spongiotrophoblast cells into the labyrinth (*Figures 3L and 4A*). Consistent with the modest increase in weight of HET-hom compared to HET-het placentas at E14.5 (*Figure 3B*), there was a 19% increase in cross-sectional area of HET-hom compared to HET-het controls (*Figure 4— figure supplement 1A*). This difference was emphasised at E17.5, with a 56% increase in HET-hom placental cross-sectional area compared to HET-het controls (*Figure 4B–C*). At E14.5, junctional zone area was significantly larger in HET-hom placentas compared to the WT-wt and HET-het controls, however, there was no difference in the HET-hom decidua area compared to all other genotypes (*Figure 4—figure supplement 1B–C*). At E17.5, the overall HET-hom placental cross-sectional area was greater than that of HET-het and WT-wt controls (*Figure 4C*), including increased total area in the junctional zone and decidua (*Figure 4—figure supplement 1D–E*). However, as a proportion of the total placental area, the junctional zone was significantly larger, resulting in a higher junctional zone/placental ratio but a similar decidua/placental ratio in HET-hom placentas compared to other genotypes (*Figure 4D–E*). As H&E and PAS staining indicated that junctional zone was expanded, we used QuPath to determine glycogen cell number and areas occupied by glycogen-enriched cells and non-glycogen cells. The total glycogen cell count was increased in HET-hom placentas but was similarly higher in both sexes (*Figure 4F*, *Figure 4—figure supplement 1F*). Moreover, the area occupied by glycogen and non-glycogen cells was increased in the decidua and junctional zones (*Figure 4G–J*). Together, while the whole HET-hom placenta was larger than controls, the fetally-derived junctional zone was disproportionately expanded and there were significantly more PAS-stained glycogen-enriched cells in midline sections of HET-hom placentas compared to all other genotypes.

As glycogen can be converted to glucose, to determine whether the greater number of glycogen-enriched cells might impact glucose levels in HET-hom offspring, we measured blood glucose levels at E17.5 and E18.5, just prior to parturition. Fetal blood glucose level was initially relatively low and similar in all genotypes at E17.5 (*Figure 4K*), but increased significantly in WT-wt, HET-wt, and HET-het controls by E18.5. However, the fetal blood glucose level in HET-hom offspring did not increase between E17.5 and E18.5 and were lower than controls at E18.5 (*Figure 4K*). This indicated that increased circulating glucose was unlikely to explain the accelerated HET-hom fetal growth observed during late gestation.

To determine whether increased fetal capillary area within the labyrinth might underlie a greater potential for maternal-fetal nutrient exchange we performed IF for CD31 (PECAM-1). Sections taken from the middle of each placenta were stained and the total labyrinth area, labyrinth/placental ratio, and fetal capillary/labyrinth ratio were quantified in WT-wt, HET-het, and HET-hom placentas (*Figure 5A–B*). In this analysis the labyrinth was defined by the CD31-positive capillary enriched region for each genotype and the total labyrinth area was defined as capillary area plus the total space outside the capillaries. The capillary area was defined as the total CD31 positive area plus the unstained space inside the capillaries. HALO analysis revealed a reduction in overall labyrinth area in both HET-het and HET-hom placentas compared to WT-wt controls (*Figure 5C*), indicating that heterozygosity for *Eed* reduced labyrinth size in the placenta. However, when total placental size was taken into account, the labyrinth/placental ratio was decreased only in HET-hom placentas compared to WT-wt and HET-het controls (*Figure 5D*). Moreover, while the capillary area to labyrinth area ratio was unchanged in HET-hom placentas compared to WT-wt controls, this ratio was reduced in HET-hom compared to HET-het placentas (*Figure 5E*). Together, these data indicate that the area available for maternal-fetal nutrient transfer in the labyrinth of HET-hom placentas was either unaffected or decreased compared to controls.

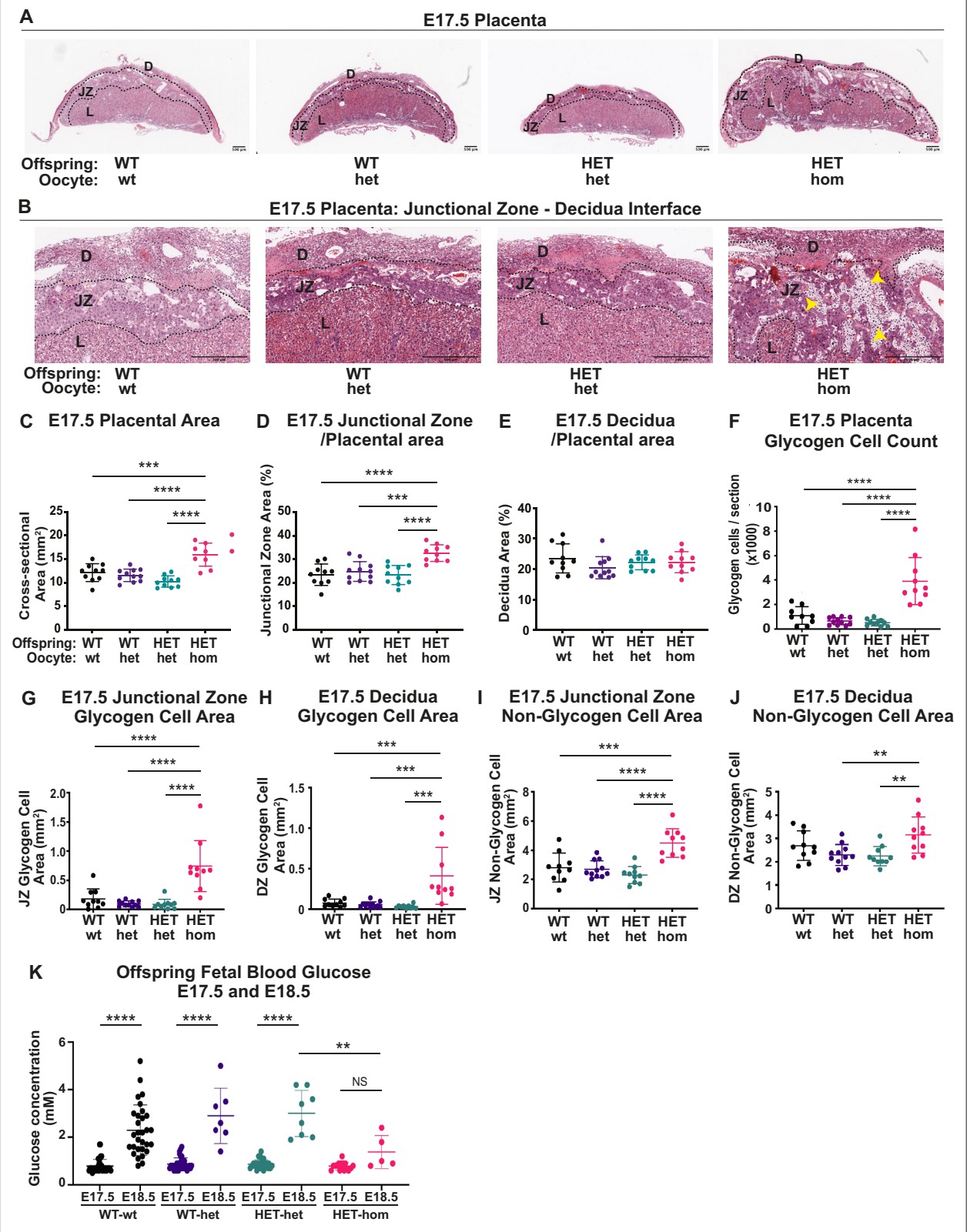

**Figure 4.** Deletion of embryonic ectoderm development (*Eed*) in the oocyte altered placental morphology in offspring. (**A,B**) Placental cross-sections at E17.5 stained with hematoxylin and eosin (H&E), black dotted line indicates the borders between the decidua (D), the junctional zone (JZ), and the labyrinth (L). Images are representative of four placentas from 10 biological replicates. Yellow arrows indicate expanded regions of glycogen cells in *Eed* HET-hom placentas. Scale bars = 500 μm. (**C–E**) Cross-sectional area of the placenta and percentage of placental cross-sectional area occupied by the

*Figure 4 continued on next page*

*Figure 4 continued*

junctional zone and decidua at E17.5. (**F**) Average number of glycogen cells/E17.5 placenta cross-section. (**G–J**) Area occupied by glycogen and non-glycogen enriched cells in the junctional zone and decidua at E17.5 For C-L: **p<0.005, ***p<0.0005, ****p<0.0001, n=10/genotype. (**K**) E17.5 and E18.5 fetal offspring blood glucose concentration. *p<0.05, n=5–30/genotype. All statistical analyses are one-way ANOVA plus Tukey's multiple comparisons. Error bars: mean ± SD.

The online version of this article includes the following figure supplement(s) for figure 4:

**Figure supplement 1.** Deletion of *Embryonic ectoderm development* (*Eed*) in the oocyte caused general placental hyperplasia, with pronounced impacts on junctional zone expansion and increased glycogen cell count.

## Widespread gene dysregulation occurs in placentas from *Eed*-hom oocytes

As the developmental changes observed in the placenta indicated that transcriptional regulation may be altered in HET-hom placentas, we analysed male and female placental tissue from E17.5 HET-hom, HET-het, and WT-wt offspring using RNA-sequencing (RNA-seq). We recovered >20 million clean, mappable reads per sample and analysed 4–6 samples for all genotypes and sexes except male HET-hom, for which only three samples were available. Principal component analysis (PCA) revealed that female HET-hom placental gene expression differed from HET-het and WT-wt placental transcriptomes (**Figure 6A**). Supporting this, differential gene expression analysis using an FDR <0.05 (false discovery rate) revealed 2083 differentially expressed genes (DEGs; together referred to as *Eed* placental DEGs) between HET-hom vs HET-het placentas (**Figure 6B**, **Supplementary file 1A**) and only four DEGs (*Eed*, *Fibin*, *Hapln4*, & *Dtx1*) between HET-het and WT-wt placentas. Together, this indicated that loss of *Eed* in oocytes rather than heterozygosity of *Eed* in offspring caused the vast majority of transcriptional dysregulation in HET-hom placentas. Gene ontology analysis indicated that the *Eed* DEGs identified between female HET-hom and HET-het placentas regulate morphogenesis, system development, and multicellular organism development (**Figure 6C**), suggesting that the gene dysregulation observed influences a diverse range of systems and processes. Surprisingly, PCA revealed that male HET-hom placental transcriptomes were similar to HET-het and WT-wt controls (**Figure 6—figure supplement 1A**), with only one sequence (CAAA01077340.1) differentially expressed between male HET-hom and HET-het placentas (FDR <0.05). However, comparison of the male and female RNA-seq data without FDR correction identified 478 male DEGs and 3964 female DEGs between HET-het and HET-hom placentas. Of these 227 were common to male and female placentas indicating that they were commonly dysregulated in male and female placentas but the impact in male placentas was potentially less pronounced (**Figure 6—figure supplement 1B**; **Supplementary file 1B-D**).

By E14.5 the majority of mature placental cell types and their key cell specific transcriptional programs are established (**Woods et al., 2018**). As single-cell RNA-seq data from E14.5 C57BL/6 placentas was publicly available (**Han et al., 2018**), we used these data to gain an indication of the placental cell types in which gene dysregulation occurred in female HET-hom placentas. Of the 2083 *Eed* placental DEGs identified in female HET-hom placentas, 543 genes were found in the E14.5 mouse placental data set (**Han et al., 2018**; **Supplementary file 1E**). Cell-specific comparisons revealed that of these 543 genes, 166 *Eed* female placental DEGs included genes preferentially expressed in the junctional zone, 57 *Eed* female placental DEGs were preferentially transcribed in the labyrinth and 104 *Eed* female placental DEGs were preferentially transcribed in the maternally derived decidua. For ease of reference, these were respectively defined as '*Eed* junctional zone DEGs,' '*Eed* labyrinth DEGs,' and '*Eed* decidua DEGs' (**Figure 6—figure supplement 2A–C**, **Supplementary file 1E**). Together these data revealed that gene dysregulation in HET-hom placentas was not isolated to a single region, and that both maternal and fetal-derived cells were altered. Of the *Eed* junctional zone DEGs 86% were upregulated, whereas 86% and 97% of the *Eed* labyrinth DEGs and *Eed* decidua DEGs were downregulated, respectively (**Figure 6—figure supplement 2D–F**, **Supplementary file 1E**). Together, these data demonstrate a strong bias for up-regulation of genes expressed in the junctional zone layer of the *Eed* placenta, which likely reflects the significant expansion observed in the junctional zone as a proportion of the total placental size.

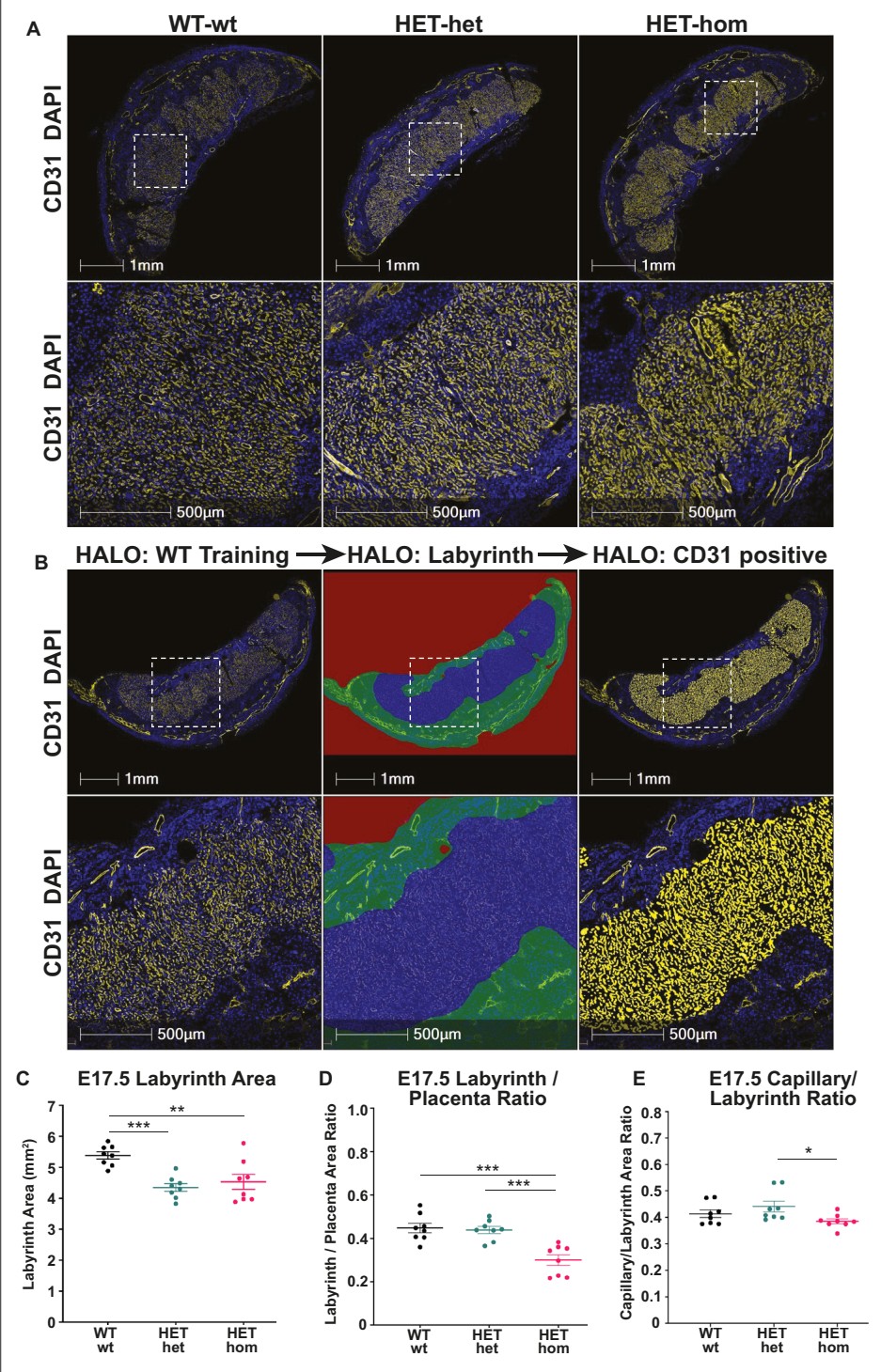

**Figure 5.** Labyrinth area and capillary/labyrinth ratios were reduced in offspring from *Embryonic ectoderm development* (*Eed*)-null oocytes. (**A**) Immunofluorescent analysis of male and female E17.5 placental sections through the middle of *Eed* WT-wt, HET-het and HET-hom placentas using CD31 to detect placental vasculature and DAPI to stain cell nuclei. (**B**) Examples of the strategy used to quantitatively analyse the total placental area and the CD31 positive staining in the labyrinth using CD31 staining (left images), the total labyrinth area, with non-capillary vasculature excluded (middle images), and the total labyrinth area and area occupied by CD31 positive capillaries (right images). (**C**) Labyrinth area; (**D**) labyrinth placenta ratio; (**E**) capillary/labyrinth ratio. One-way ANOVA plus Tukey's multiple comparisons. *$p<0.05$, **$p<0.005$, ***$p<0.001$, n=8 WT-wt (four male and four female; nine and eight sections), eight HET-het (four male and four female; 9 and 12 sections) and 8 HET-hom (four males and four femalse; 16 and 8 sections) placentas. Error bars: mean ± SD.

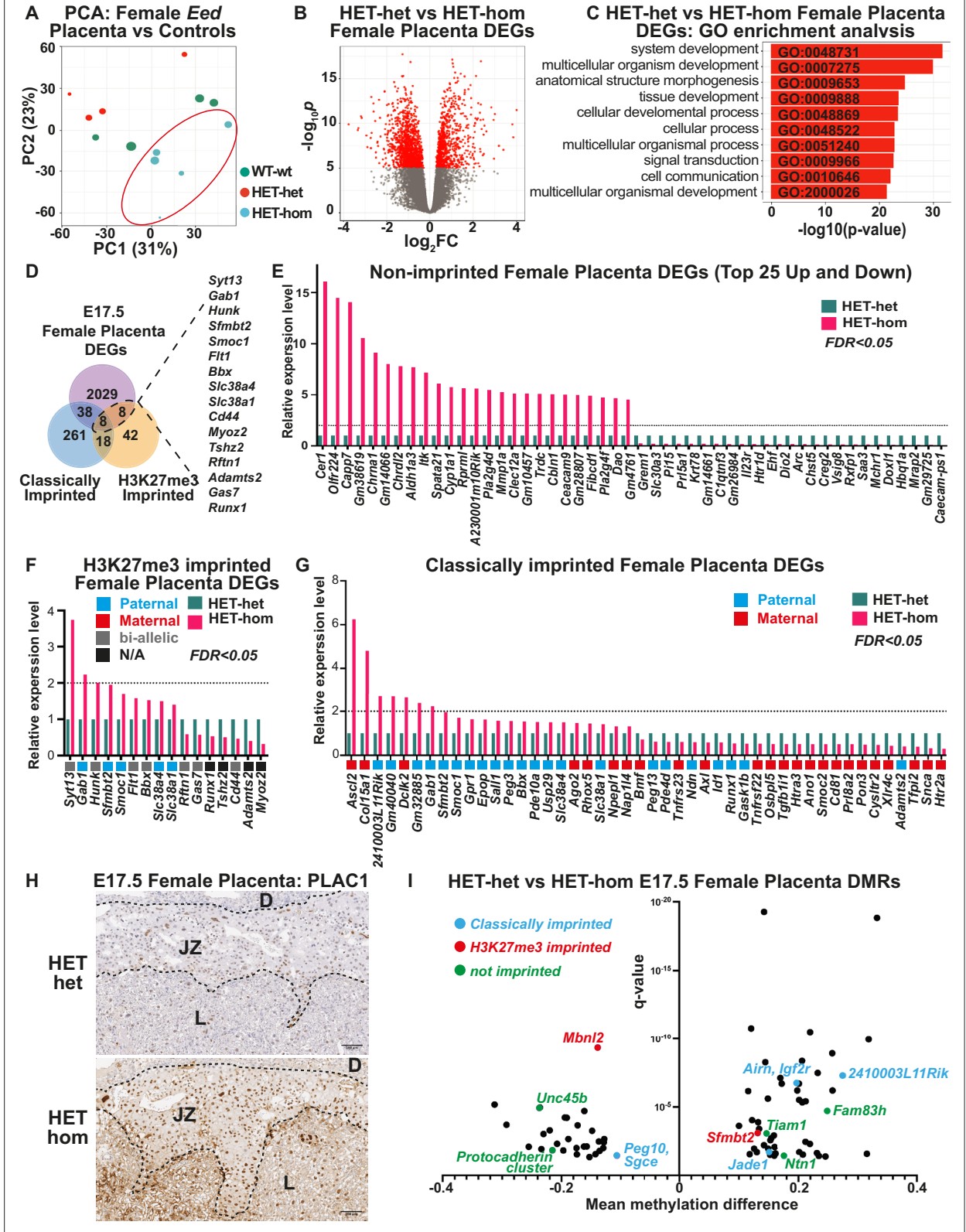

**Figure 6.** Loss of Embryonic ectoderm developmen (EED)-dependent oocyte programming altered placental transcription. (**A**) Principal Component Analysis (PCA) of E17.5 female *Eed* HET-hom, HET-het, and WT-wt placenta bulk RNA-seq data. n=4–5/genotype. (**B**) Differential gene expression analysis of E17.5 HET-het vs HET-hom placenta represented by a volcano plot showing logFC against statistical significance. Genes with false discovery rate (FDR)-adjusted FDR <0.05 are coloured in red. Deletion of *Eed* in the oocyte resulted in 2083 differentially expressed genes (DEGs) in HET-hom

*Figure 6 continued on next page*

*Figure 6 continued*

placenta (*Eed* placental DEGs). (**C**) GO enrichment analysis of HET-het vs HET-hom DEGs representing the top 10 significantly different biological processes impacted. (**D**) Venn analysis of DEGs identified between E17.5 female HET-het and HET-hom placentas showing non-imprinted, classically imprinted, and H3K27me3 imprinted genes. (**E–G**) Relative transcription levels of female E17.5 HET-hom placental DEGs including (**E**) the top 25 non-imprinted genes that were higher and lower in HET-hom vs HET-het samples, (**F**) H3K27me3 imprinted genes identified (**G**) Classically imprinted genes identified. Only DEGs with significant expression differences and an FDR <0.05 are shown. In (**F**) and (**G**) the typical maternal (red) /paternal (blue)/bi-allelic (gray) expression pattern is indicated below each graph. The dotted black line shows the level for a twofold increase in expression. (**H**) Representative immunohistochemical analysis of PLAC1 (encoded by the X-linked gene *Plac1*) in mid-cross sections of E17.5 HET-het and HET-hom placentas. The black dotted lines show boundaries between the decidua (D), the junctional zone (JZ), and the labyrinth (L). n=4/genotype. Scale bars = 500 um (top), 100 um (bottom). (**I**) Mean methylation difference vs significance for Differentially methylated CpG-rich regions (DMRs) in E17.5 female HET-het vs HET-hom placentas with DMRs within 1 kb of classically imprinted (blue), histone 3 Lysine 27 trimethylation (H3K27me3)-imprinted (red), and differentially expressed non-imprinted genes (green) shown.

The online version of this article includes the following figure supplement(s) for figure 6:

**Figure supplement 1.** Transcriptional analysis using a confidence level of p<0.05 revealed commonly altered genes between E17.5 female vs male *Eed* HET-hom placentas.

**Figure supplement 2.** Comparison of female placenta differentially expressed genes (DEGs) revealed differential transcriptional impacts in the junctional zone, labyrinth, and decidua in *Eed* HET-hom placentas.

**Figure supplement 3.** Loss of Embryonic ectoderm development (EED) in the oocyte increased PLAC1 expression in the developing placenta, but did not result in overrepresentation for increased expression of X-linked genes.

## *Eed* placental DEGs included non-imprinted genes, H3K27me3 and classically imprinted genes, and a small number of X-linked genes

The majority (2029/2083; 97.4%) of placental DEGs were bi-allelically expressed autosomal genes that are associated with a range of developmental processes (*Figure 6C–E*). In addition, of 76 candidate non-canonical H3K27me3 imprinted genes identified by *Inoue et al., 2017*, 16 were dysregulated in HET-hom vs HET-het placentas, though eight of these genes were also listed as classically imprinted genes in the list we examined (*Figure 6C and F*). Notably, *Slc38a4*, *Slc38a1*, *Sfmbt2*, *Gab1,* and *Smoc1* are normally paternally expressed (maternally imprinted/silenced), but were upregulated in female HET-hom placentas consistent with de-repression of the maternal allele (*Figure 6F*, *Supplementary file 1F*). Loss of H3K27me3-dependent imprinting at these genes has been functionally linked to placental hyperplasia, including in offspring derived from SCNT (*Inoue et al., 2020*; *Wang et al., 2020*; *Xie et al., 2022*; *Matoba et al., 2022*). Given that both *Eed* HET-hom offspring and SCNT offspring lack H3K27me3 imprinting, we compared E14.5 and E19.5 fetal and placental weight data collected in this study with SCNT offspring data extracted from *Matoba et al., 2018*. This revealed very similar fetal and placental growth trajectories in the *Eed* HET-hom and SCNT offspring models (*Figure 7*).

Of the remaining DEGs listed as H3K27me3 imprinted genes (*Figure 6F*), only *Slc38a1* is usually paternally expressed. The remainder are either maternally or bi-allelically expressed indicating that they are unlikely to be regulated by H3K27me3 imprinting in oocytes. In addition to H3K27me3 imprinted genes, classically imprinted genes were also dysregulated in HET-hom placentas. Of 325 candidate or confirmed classically imprinted genes we examined (*Andergassen et al., 2017*; *Wani-gasuriya et al., 2020*), 22 were increased in expression including 15 paternally expressed (maternally imprinted/silenced) and seven maternally expressed (paternally imprinted/silenced) genes (*Figure 6G*, *Supplementary file 1F*). Three (*Gab1*, *Sfmbt2*, and *Smoc1*) of the upregulated paternally expressed genes were also identified in our analysis of H3K27me3 imprinted genes. Similarly, of 24 genes that were decreased in expression, 17 are normally maternally expressed and seven are paternally expressed genes (*Figure 6G*, *Supplementary file 1F*).

Maternal EED is required for regulating X-inactivation in pre-implantation embryos and loss of EED in oocytes causes male-biased HET-hom fetal lethality (*Figure 1H*; *Inoue et al., 2018*; *Harris et al., 2019*). Moreover, loss of H3K27me3 imprinting increases maternal *Xist* expression and deletion of maternal *Xist* partially rescued growth delay caused by loss of EED in oocytes (*Matoba et al., 2022*). *Xist* was not significantly increased in HET-hom placentas in this study. However, the X-linked genes *Plac1* and *Ldoc1* were increased in female HET-hom compared to HET-het placentas but were not differentially expressed in male placentas raising the possibility that these genes escape X-inactivation in Het-hom placentas (*Supplementary file 1B-D*). In line with its increased transcription, PLAC1

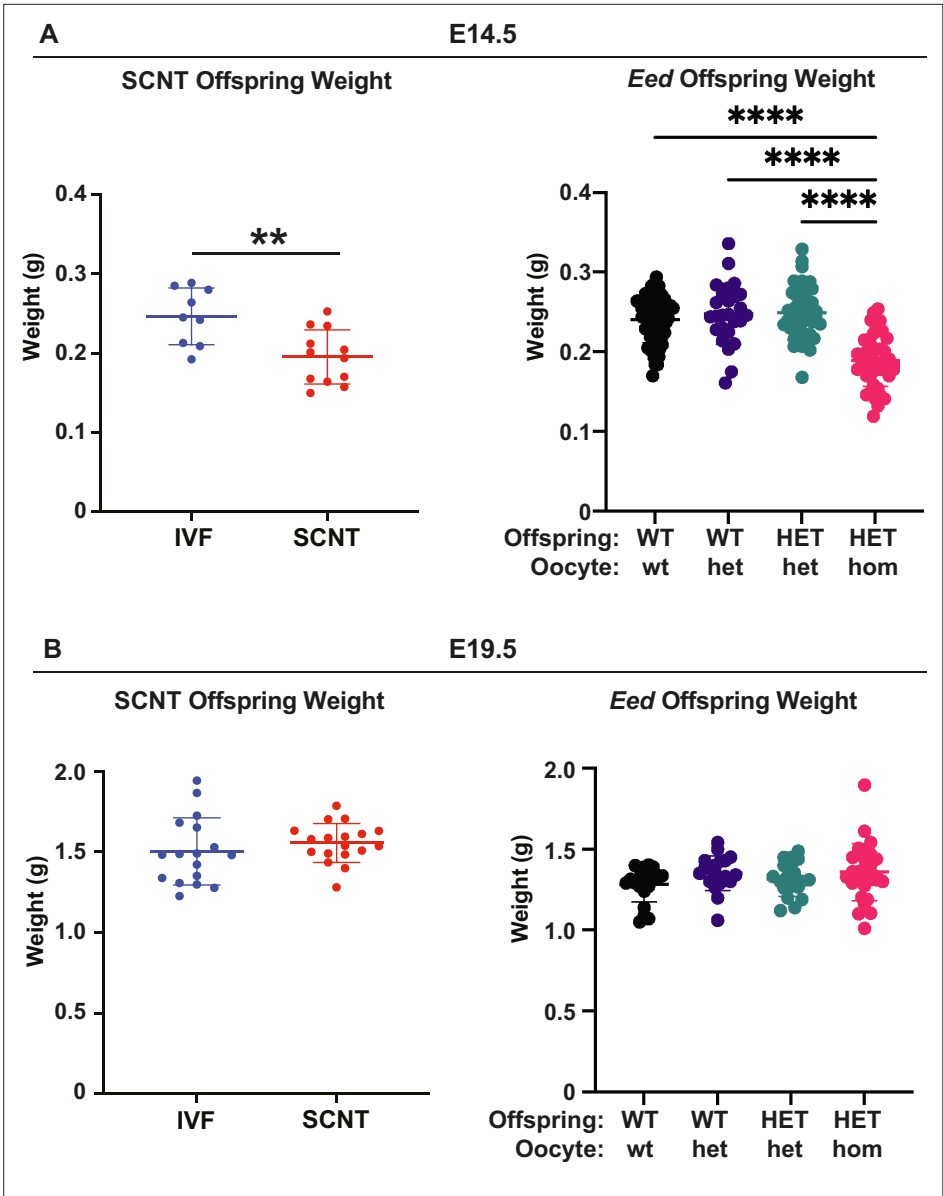

**Figure 7.** *Eed* HET-hom offspring demonstrate similar growth profiles to somatic cell nuclear transfer (SCNT) mice. Comparison of fetal weights of SCNT mice from Xie et al., (*Xie et al., 2022*) and offspring from the *Eed-ZP3*-Cre mouse model at (**A**) E14.5 and (**B**) E19.5. N values - E14.5: IVF 9; SCNT 12; *Eed* WT-wt 64; WT-het 26; HET-het 45; HET-hom 43. E19.5 IVF 17; SCNT 18; *Eed* WT-wt 21; WT-het 20; HET-het 19; HET-hom 28. **p<0.005, ****p<0.0001. One-way ANOVA with Tukey's multiple comparisons. Error bars: mean ± SD.

protein was also increased in cells within the junctional zone (*Figure 6H*, *Figure 6—figure supplement 3*). These data are of interest as together with the imprinted genes *Ascl2* and *Peg3*, *Plac1*, and *Ldoc1* are known to regulate placental glycogen stores (*Tunster et al., 2020*) and have been associated with junctional zone development (*Tunster et al., 2016*; *Tunster et al., 2018*; *Lee et al., 2015*; *Allas et al., 2019*; *Thiaville et al., 2013*; *Naruse et al., 2014*). Despite the increased expression of this subset of X-linked genes, placental DEGs occurred at a similar rate on the X-chromosome (p=0.26626) and autosomes (*Supplementary file 1G*). Moreover, while there were 81 X-linked DEGs, 32 had increased and 49 had reduced expression (*Supplementary file 1A*) indicating that there was no bias for increased expression of X-linked genes in HET-hom placentas. However, there was a significant overrepresentation of genes located on chromosomes 6 (p=0.000375) and 15 (p=0.03088) genes encoded by the mitochondrial genome (p=0.00202) in the DEG list (*Supplementary file 1G*).

Included in the mitochondrial sequences were *mt-Co1, mt-Co2,* and *mt-Co3* (mitochondrial cytochrome oxidase C I-III), *mt-Nd3, mt-Nd4,* and *mt-Nd4l* (mitochondrial NADH dehydrogenase 3 and 4), and *mt-Atp6* (mitochondrial ATP synthase 6). These are of interest given that mitochondrial function is dysregulated in placentas with compromised function (*Colson et al., 2021*).

## DNA methylation changes in E17.5 placentas are found at a subset of DEGs

While loss of EED in oocytes disrupts H3K27me3 imprints (*Hanna and Kelsey, 2021*; *Inoue et al., 2017*), our data suggest that other genes may also be affected, perhaps through altered DNA methylation (*Jarred et al., 2022*). We, therefore, analysed DNA methylation in placentas of E17.5 female WT-wt, HET-het, and HET-hom offspring using RRBS. This identified 81 DMRs between HET-het and HET-hom placentas, 87 DMRs between WT-wt and HET-hom placentas, and 27 DMRs between WT-wt and HET-het placentas (FDR <0.05; *Figure 6I*; *Supplementary file 1H-J*). Of the 81 HET-het vs HET-hom DMRs, two were located within or nearby the H3K27me3 imprinted genes *Sfmbt2* and *Mbnl2*. The DMR associated with *Sfmbt2* was 25% methylated in the HET-het placentas but 10% methylated in the HET-hom group (*Supplementary file 1H*), which may explain its increased transcript levels in HET-hom placentas. In addition, of the 81 HET-het vs HET-hom placental DMRs, seven were within 1 kb of the orthologous human region that had H3K27me3 peaks in human oocytes (*Xia et al., 2019*), including one 201 bp upstream of the *SFMBT2* transcriptional start site.

Also included in the 81 HET-het vs HET-hom placental DMRs were four within or nearby the classically imprinted genes *Igf2r, Airn, Peg10, Sgce, Jade1,* and *2410003L11Rik*, and 74 DMRs at non-imprinted loci. The latter included five DMRs within or nearby eight non-imprinted genes (*Ntn1, Unc45b, Fam83h, Tiam1, Pcdhga1, Pcdhga5, Pcdhgb5,* and *Pcdhgb7*) that were differentially expressed between HET-het and HET-hom placentas (*Figure 6I*; *Supplementary file 1G*). Of these, *NTN1* (Netrin1) is of particular interest as it is reduced in the placentas of women with fetal growth restriction and potentially influences placental size by increasing the viability of placental microvascular endothelial cells (*Qian-hua et al., 2011*; *Wang et al., 2011*). While *Ntn1* is not imprinted, it had reduced DNA methylation, consistent with its increased transcription in HET-hom compared to HET-het placentas (*Supplementary file 1H*).

## Loss of EED in the oocyte did not alter late gestational fetal blood metabolite levels

A previous study of SCNT-derived mice indicated that loss of non-canonical imprinting increased *Slc38a4* expression, resulting in placental hypertrophy and increased amino acid transport (*Xie et al., 2022*). To determine whether altered placental function changed fetal blood amino acid levels and/or fetal or maternal metabolomic state, we collected blood samples from male and female E17.5 WT-wt, HET-het, and HET-hom fetal offspring and their mothers and assessed 356 metabolites using mass spectrometry. One hundred and sixty metabolites were detected across the sample sets. Principal component analysis revealed three separate clusters corresponding to maternal and fetal blood samples and 19 pooled biological quality control samples (PBQCs: included to ensure data reproducibility), demonstrating sensitivity to detect metabolic differences (*Figure 8A*). The maternal samples clustered separately from the fetal samples, revealing clear metabolic differences between the fetal and maternal bloods. However, HET-hom and WT-wt or HET-het of both sexes were inseparable using PCA (*Figure 8B*) and there were no significant differences that were consistent between HET-hom and WT-wt or HET-het fetal blood samples of either sex (FDR <0.05; *Figure 8C*). For example, while 14 metabolites (10 with a mean difference >0.5) were statistically significant between female HET-hom and HET-het fetal blood samples (*Figure 8C*, *Supplementary file 1K*), none of the same metabolites differed between female HET-hom and WT-wt samples (*Figure 8C*, *Supplementary file 1L-M*) and there were no differences between male HET-hom and HET-het samples (FDR <0.05; *Figure 8C*). Given we expect similar changes between HET-het and WT-wt controls compared to HET-hom samples, we concluded that the differences detected were unlikely to be due to altered placental function in HET-hom offspring. Moreover, although differences in levels of several amino acids have been reported in fetal blood samples of SCNT-derived animals (*Xie et al., 2022*), we detected no consistent differences in amino acids in HET-hom fetal blood samples and their counterpart HET-het and WT-wt controls in

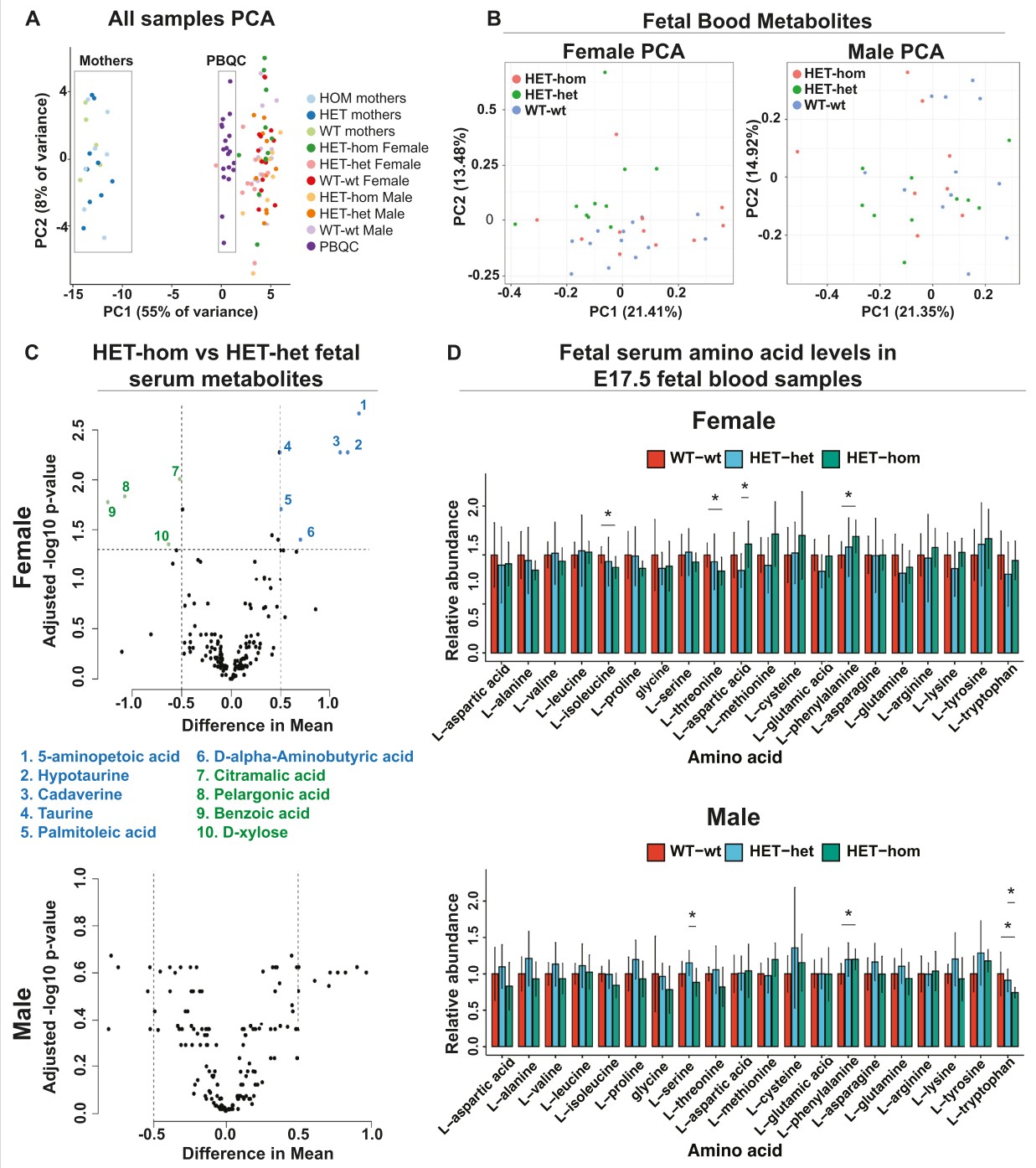

**Figure 8.** Loss of Embryonic ectoderm development (EED) in the oocyte did not overtly affect the fetal blood metabolomic state in late gestation offspring. (**A**) Principal component analysis (PCA) of male and female E17.5 fetal serum samples, matched maternal serum samples, and pooled serum quality controls (**B**) PCA plots for metabolites in male and female fetal serum samples from *Eed* WT-wt, HET-het and HET-hom offspring. (**C**) Volcano plots showing significant differences and difference in mean levels for metabolites and relative amino acid levels in male and female HET-hom vs HET-het fetal serum samples (**D**) Relative serum levels for all amino acids in female and male WT-wt, HET-het, and HET-hom offspring. Students *t*-test. *p<0.05; N=8–12. Error bars: mean ± SD. (**A–D**) Total sample set: 85 samples: n=8–12 for fetal serum samples (Female: 12 WT-wt; 10 HET-het; 12 HET-hom. Male: 12 WT-wt; 10 HET-het; 8 HET-hom). n=5–9 for maternal serum samples (5 wt; 9 het; 7 hom). Of 356 metabolites tested, 160 were reliably detected in the fetal serum samples.

either sex (*Figure 8D*). Phenylalanine was modestly increased in female and male HET-hom vs WT-wt samples, but was unchanged between HET-hom and HET-het samples of either sex (*Figure 8D*).

Finally, serum metabolites of mothers carrying WT-wt, WT-het/HET-het, and HET-hom pregnancies clustered together. Moreover, no significant differences in metabolites were detected between the genotypes indicating that the blood metabolomes of these females were similar late in pregnancy, despite the substantial differences in placental and fetal growth profiles of the HET-hom offspring (*Figure 8A*). Together, these data indicate that an increased supply of amino acids or other metabolites across the placenta is unlikely to account for the accelerated growth we observed in late gestational HET-hom offspring.

## Discussion

Fetal growth restriction is commonly caused by placental insufficiency investigated using surgical or nutritional interventions in animal models (*Palliser et al., 2010*; *Louey et al., 2000*; *Swanson and David, 2015*; *Vuguin, 2007*). In this study similar outcomes were caused by deleting *Eed* in the oocyte, resulting in the production of isogenic HET-hom offspring that are epigenetically different to HET-het controls. Initially, offspring were characterised by embryonic and fetal developmental delay, but this was resolved late in gestation. Despite lower placental efficiency indicated by fetal/placental weight ratio, low fetal glucose levels, and unaffected amino acid and metabolomic profiles in late gestation fetal and maternal serum, HET-hom offspring underwent a period of late fetal growth recovery and early postnatal overgrowth. Although placental transcription and DNA methylation was altered at important H3K27me3-imprinted genes and some classically imprinted genes, transcription and DNA methylation were also altered at many non-imprinted sites. In addition, several X-linked genes that have been associated with placental development and function were transcriptionally increased, but we found no evidence for widespread dysregulation of X-inactivation in late-stage placentas. While the underlying cause remains unclear, our data indicate that fetal growth is normalised in HET-hom offspring in the absence of any obvious increase in nutritional support from the placenta and that loss of EED in oocytes causes programming effects on the fetus that are unlikely to be explained by loss of only H3K37me3-dependent imprinting or increased maternal *Xist*.

A previous study reported that blastocysts from *Eed*-null oocytes formed at normal rates and were not affected by increased cell death, indicating normal pre-implantation development (*Inoue et al., 2018*). However, we found that while preimplantation development progressed at similar rates in embryos from *Eed* wild-type and *Eed*-null oocytes, blastocysts derived from *Eed*-null oocytes contained low cell numbers, primarily due to fewer inner cell mass cells. This deficit may be due to low cell proliferation during pre-implantation development caused by loss of EED-dependent oocyte programming or to loss of maternal EED supplied in the mature oocyte (*Inoue et al., 2017*; *Inoue et al., 2018*; *Jarred et al., 2022*; *Harris et al., 2019*; *Erhardt et al., 2003*). While the latter is consistent with the established role of PRC2 in driving cell division in stem cells and other cell types (*Bracken et al., 2003*), other EED-dependent effects in the oocyte may also contribute.

Consistent with previous studies (*Matoba et al., 2018*; *Xie et al., 2022*; *Matoba et al., 2022*), later development of offspring generated from oocytes lacking EED was characterised by developmental delay of the fetus and placental hyperplasia. Placental hyperplasia results from loss of H3K27me3 imprinting in SCNT and from maternal deletion of *Eed* (*Matoba et al., 2018*; *Xie et al., 2022*; *Matoba et al., 2022*), demonstrating a common impact on placental development in both models. Moreover, comparison of our data with SCNT offspring (*Matoba et al., 2018*) indicated that fetal growth restriction and subsequent fetal growth recovery occur in *Eed*-null oocyte and SCNT-derived offspring, and that this phenotype was resolved during pregnancy. While both models involve loss of H3K27me3-dependent imprinting (*Inoue et al., 2018*; *Wang et al., 2020*; *Xie et al., 2022*), the HET-hom offspring in this study were derived from oocytes that lacked maternal *Eed*/EED (RNA or protein) and SCNT embryos were derived from wild-type enucleated oocytes that presumably contained maternal PRC2. Therefore, while the lack of maternal EED could explain the lower number of cells observed in *Eed* HET-hom preimplantation offspring, this is less likely in SCNT embryos. However, early development of SCNT embryos is restricted by the low efficiency of oocyte-driven reprogramming (*Wang et al., 2020*; *Matoba et al., 2014*; *Liu et al., 2016*; *Cao et al., 2013*), and comparison of the earliest stages of development between these models is challenging. Despite this, growth and developmental delay appear to be resolved late in gestation in both models without correction of H3K27me3-imprinting,

indicating a level of plasticity that permits delivery of fully grown pups. Even though prolonged gestation decreased pup survival in the *Eed* model, an interesting conjecture may be that an unknown sensing mechanism in these growth-restricted pregnancies supports adaptation(s) that lead to late fetal growth and extended gestation, favouring the delivery of viable pups. While speculative, such a mechanism could involve epigenetic adaptation during early or late development and/or altered epigenetic control of imprinted or non-imprinted genes in the placenta and/or fetus.

Placental hyperplasia in HET-hom offspring was characterised by an increased number of glycogen-enriched cells and increased junctional zone and decidua area occupied by glycogen-enriched cells and non-glycogen cells. While increased size was observed particularly in the fetally-derived junctional zone, the maternal side of the placenta also became larger, perhaps facilitating overall tissue balance in the placenta in response to the hyperplastic junctional zone. The number of glycogen-enriched trophoblasts usually peaks at E16.5 and declines by approximately 60% by E18.5 (*Tunster et al., 2020*), potentially releasing glycogen stores to the mother and/or fetus. Consistent with this, fetal blood glucose levels increased between E17.5 and E18.5 in WT-wt, WT-het, and HET-het control offspring. However, despite the increased numbers of glycogen-enriched cells in HET-hom placentas, fetal blood glucose levels did not significantly increase in HET-hom offspring. Moreover, the decreased fetal/placental weight ratio observed in HET-hom offspring suggested decreased placental efficiency and we did not observe any increase in labyrinth area, or the fetal labyrinth/placenta or capillary/labyrinth ratios, suggesting that the area devoted to maternal-fetal nutrient exchange was not substantially affected. Consistent with this, we found no change in levels of fetal blood amino acids or other metabolites, also suggesting that placental function was not obviously enhanced. Together, increased maternal-fetal placental exchange, glucose, or amino acid supply seem unlikely to explain the ability of substantially growth-restricted HET-hom offspring to attain normal weight by birth, indicating that unknown mechanism(s) contribute.

In addition to facilitating increased glucose release and fetal growth, glycogen-producing trophoblasts are considered to inhibit the release of placental factors such as oxytocin and prolactins, which prepare the pregnancy for parturition (*Harris et al., 2019*; *Lee et al., 2015*). Therefore, one possibility may be that the increased glycogen cell number in late-gestation HET-hom placentas inhibits the release of oxytocin, prolactins, or other hormones, causing the delay in parturition observed in 60% of the pregnancies from *Eed*-hom oocytes. Moreover, while extended gestation may also allow additional time for growth delay to resolve, HET-hom pup mortality increased in pregnancies extended by one or two days, indicating that there is a substantial cost for lengthening pregnancy in this model.

As well as morphological alterations observed in HET-hom placentas, we found extensive transcriptional dysregulation with 2083 DEGs identified between HET-hom and HET-het placentas. Consistent with the increased size of fetal and maternal tissues in HET-hom placentas, comparison of these DEGs with single-cell data from E14.5 normal placentas (*Han et al., 2018*) indicated that altered gene expression was not restricted to genes normally expressed in a single region of the HET-hom placenta, but affected fetal and maternal layers. While one might expect oocyte-specific loss of EED to affect only fetally-derived placental tissue, this ignores the possibility that increased size of the maternal tissue may occur in response to hyperplasia in the fetally-derived layers. Indeed, it seems plausible that the placenta may not properly support late fetal growth without remodelling maternally-derived placental tissue and inducing associated transcriptional change.

Another study demonstrated that loss of H3K27me3 imprinting of *Xist* contributes to the fetal developmental delay and male-biased fetal death of offspring derived from *Eed*-null oocytes (*Matoba et al., 2022*). Oocyte-specific deletion of both *Xist* and *Eed* increased survival from approximately 3–6 pups/litter, but a significant deficit in litter size remained compared to the average of ~8.5 pups in wild-type control litters. Moreover, while *Xist/Eed* maternal double deletion improved fetal body weight, it remained marginally, but significantly lower than wild-type control. Therefore, while deletion of maternal *Xist* rescued male-biased fetal death observed as a result of *Eed* deletion in oocytes, it did not completely restore offspring growth or survival, demonstrating that other mechanisms contribute (*Matoba et al., 2022*). Moreover, the fetal growth recovery of HET-hom offspring described in our study cannot be explained by loss of *Xist*, indicating that the mechanism involved in the late gestational resolution of HET-hom fetal growth and development remains unresolved.

The DEGs identified in HET-hom hyperplastic placentas included increased expression of H3K27me3 imprinted genes *Slc38a4, Sfmbt2, Gab1, and Smoc1,* nine classically imprinted genes, X-linked genes

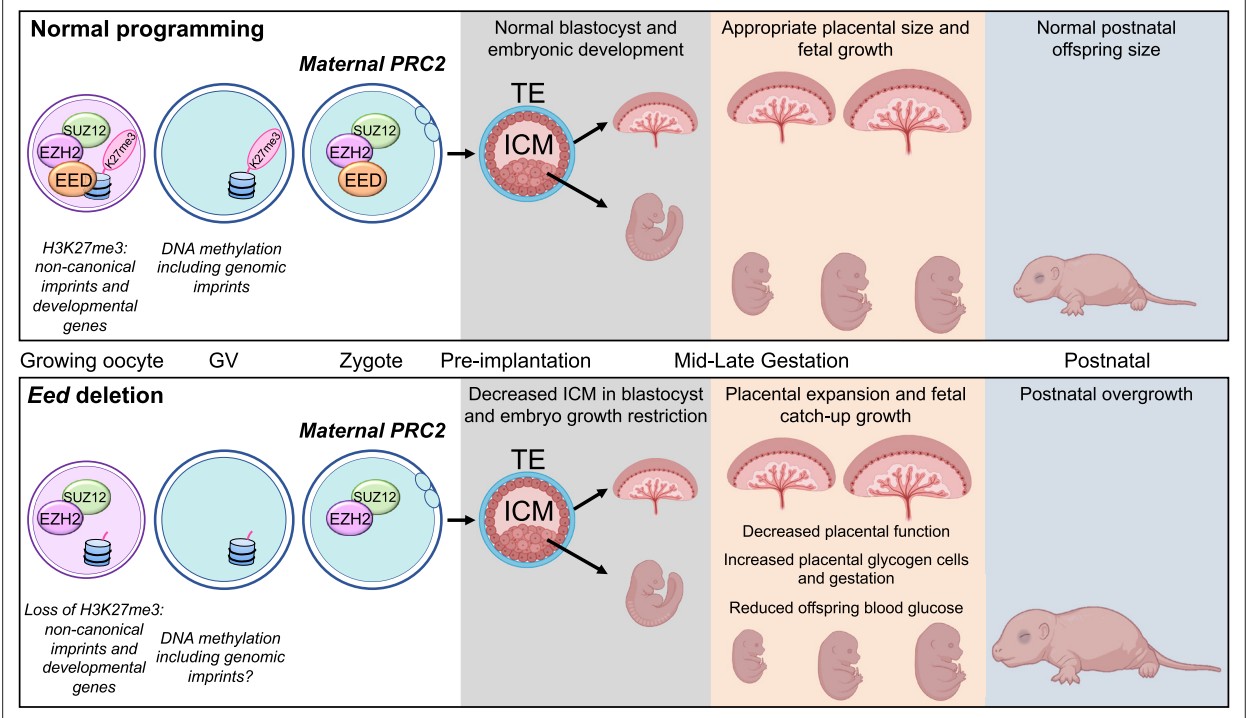

**Figure 9.** Summary of the placental and offspring growth response due to Embryonic ectoderm development (EED) loss in the oocyte. During oocyte growth, all three subunits of polycomb repressive complex 2 (PRC2) are present at the primary to secondary stages, which is important for the silencing of developmental genes and Histone 3 Lysine 27 trimethylation (H3K27me3)-dependent imprinting (**Inoue et al., 2017**; **Inoue et al., 2018**; **Matoba et al., 2022**; **Jarred et al., 2022**). Deletion of *Eed* in oocyte results in loss of maternal H3K27me3 and PRC2, early growth restriction, followed by placental hyperplasia and late gestation fetal catch-up growth, outcomes consistent with loss of H3K27me3-dependent imprinting observed in SCNT offspring (**Matoba et al., 2018**; **Inoue et al., 2020**; **Wang et al., 2020**). Placentas generated from oocytes lacking EED have expanded glycogen-enriched cells in the junctional zone and significant gene dysregulation in the placenta. Despite placental hyperplasia and reduced placental efficiency late in gestation, offspring catch-up growth observed in this model may be explained by loss of imprinting for *Slc38a4*, increased placental amino acid transport, and extended gestational length, explaining why these offspring are overgrown immediately after birth (**Prokopuk et al., 2018**).

including *Plac1*, *Wdr1*, and *Ldoc1* and many bi-allelically expressed autosomal genes. Consistent with increased *Plac1* transcription, immunohistochemistry confirmed that PLAC1 expression was increased in junctional zone cells in HET-hom placentas. However, rather than increased expression, deletion of *Plac1* has been associated with male-biased lethality and a similar placental phenotype to that observed in HET-hom offspring (**Jackman et al., 2012**). Notwithstanding this difference, it seems likely that placental hyperplasia in HET-hom offspring could involve *Plac1*, *Wdr1*, *Ldoc1*, and/or autosomal genes, perhaps in conjunction with *Slc38a4* or *Sfmbt2/C2MC*, which clearly contribute to placental hyperplasia caused by loss of EED in the oocyte (**Matoba et al., 2022**).

Comparison of HET-hom with HET-het placentas using RRBS identified 81 DMRs, including H3K27me3 imprinted loci *Sfmbt2/C2MC* and *Mbnl2*, five classically imprinted loci and 74 non-imprinted loci, indicating that loss of EED in oocytes affects epigenetic status at both imprinted and non-imprinted genes. However, while HET-hom and HET-het offspring are isogenic and these data demonstrate that DNA methylation changes occur in the placenta as a result of deletion of *Eed* in oocytes, it is unclear that this is a direct effect. Moreover, as we observed 27 DMRs between HET-het and WT-wt placentas heterozygosity for *Eed* may also contribute, although only four differentially expressed genes, including *Eed*, were identified in the same comparison. This supports the idea that perturbation of H3K27me3 in the oocyte leads to altered DNA methylation and transcription of imprinted and non-imprinted genes that are influential in placental function and embryo growth. However, further work is required to determine the extent to which epigenetic changes in the oocyte directly affect embryo and placental development in offspring.

In summary, we demonstrate that loss of PRC2 function in the oocyte results initially in embryonic developmental delay and fetal growth restriction, followed by placental hyperplasia and late fetal

growth that results in normalisation of offspring weight at birth and perinatal overgrowth (*Figure 9*). Moreover, it appears that offspring derived from either *Eed*-null oocytes or by SCNT have an innate ability to correct fetal growth restriction during the late stages of pregnancy despite maternally inherited impacts mediated by *Xist* or H3K27me3-dependent imprinting. In offspring from *Eed*-null oocytes normalisation of fetal growth occurs despite low placental efficiency and in the absence of enhanced glucose, amino acid, and metabolite levels, indicating that unknown mechanisms contribute. Together, this work reveals that altered PRC2-dependent programming in the oocyte elicits a complex intra-uterine response that supports compensatory fetal growth despite apparently negative impacts on placental development and function. As fetal growth restriction and fetal catch-up growth have been linked with negative health outcomes later in life, including metabolic conditions (*Singhal, 2017*; *Singhal and Lucas, 2004*), this model may provide opportunities for understanding the physiological basis of such outcomes. Moreover, given that mutations in *EED, EZH2,* and *SUZ12* have all been associated with overgrowth and a range of co-morbidities in Cohen-Gibson, Weaver, and Imagawa-Matsumoto syndrome patients (*Cohen and Gibson, 2016*; *Tatton-Brown et al., 2011*; *Cooney et al., 2017*; *Imagawa et al., 2017*; *Tatton-Brown et al., 2013*), and similar outcomes have been observed in mouse offspring lacking EED in oocytes (*Prokopuk et al., 2018*), further work may provide insights into these rare human conditions.

## Methods
### Mouse strains, animal care, and ethics
Mice were housed using a 12 hr light-dark cycle at Monash Medical Centre Animal Facility, as previously reported (*Prokopuk et al., 2018*). Room temperature was maintained at 21–23°C with controlled humidity, and food and water were provided ad libitum. All animal work was undertaken in accordance with Monash University Animal Ethics Committee (AEC) approvals. Mice were obtained from the following sources: *Zp3Cre* mice C57BL/6-Tg 93knw/J; Jackson Labs line 003651, constructed and shared by Professor Barbara Knowles (*de Vries et al., 2000*), *Eed* floxed mice (*Eed*<sup>fl/fl</sup>) B6; 129S1-*Eed*tm1Sho/J; Jackson Labs line 0022727; constructed and shared by Professor Stuart Orkin (*Yu et al., 2009*). The *Eed* line was backcrossed to a pure C57BL6/J and shared with us by Associate Professor Rhys Allen and Professor Marnie Blewitt, Walter and Eliza Hall Institute for Medical Research, Melbourne.

### Genotyping
Genotyping was performed by Transnetyx (Cordova, TN) using real-time PCR assays (details available upon request) designed for each gene as described previously (*Prokopuk et al., 2018*).

### Collection and culture of pre-implantation embryos
Eight to twelve-week-old female mice were superovulated and mated to C57BL/6 males for one night. Zygotes were collected in handling media (G-MOPS PLUS, Vitrolife) at 37 °C (*Gardner and Lane, 2014*; *Gardner and Truong, 2019*) denuded of cumulus cells with G-MOPS PLUS containing hyaluronidase. All embryos were washed in G-MOPS PLUS and embryo development kinetics was assessed using the EmbryoScope (Vitrolife) time-lapse imaging system. Embryos were cultured individually in 25 µl of medium, with time-lapse images generated at 15 minute (min) intervals throughout the culture period.

### Cell allocation in blastocysts
Following EmbryoScope culture, differential staining was performed (*Hardy et al., 1989*) in hatched blastocysts using propidium iodide to label TE nuclei, while leaving the ICM unlabelled. After fixation, embryos were treated with bisbenzimide to stain ICM and TE, whole-mounted in glycerol, and imaged using an inverted fluorescence microscope (Nikon Eclipse TS100). Nuclei were counted using ImageJ.

### Collection of post-implantation embryos, placenta, and postnatal offspring
Mice were time mated for two-four nights, with females plug checked daily for copulation plugs. Positive plugs were noted as day E0.5 and all females for which a plug was discovered were immediately

separated from the male. Gestational length was measured in days post copulation by recording the morning of a copulation plug was detected as E0.5 and visually monitoring females twice daily (morning and afternoon) for births from late gestation (E18.5) until pups were delivered (E19.5-E21.5, depending on oocyte genotype). Pregnant females were euthanised and embryos were collected at E9.5, E12.5 E14.5, E17.5, and E18.5. E9.5 whole sacs were weighed, fixed in 4% PFA for 72 hr at 4 °C, processed, paraffin-embedded, and sectioned at 5 μm. Whole-mount images of E9.5 embryos were taken using a LEICA M80 light microscope with LEICA MC170 HD camera attachment. Embryos and placentas were collected from the same offspring at E12.5, E14.5, E17.5, and E18.5 and weighed separately. Placentas were then bisected, and half was fixed in 4% PFA for 72 hr at 4 °C, processed, and paraffin-embedded with the cut side of the placenta facing the front of the block. The other half of the placenta was rinsed in PBS, snap-frozen on dry ice, and stored at –80 °C for RNA analysis and RRBS. P0 (day of birth) or P3 pups were weighed, euthanised by decapitation, and samples collected as required.

## Placental histology

Each block was trimmed to the minimum extent possible to allow collection of full placental sections. Blocks were sectioned in compound series at 5 μm using a Leica microtome and sections transferred to Superfrost plus slides (Thermo Fisher Scientific). With as much accuracy as possible, every section was collected, starting from the midline of each placenta. Two slides at the start of the series were used for H&E and PAS staining and slides from the remaining set were used for PLAC1 immunohistochemistry and CD31 immunofluorescence. Periodic antigen-shiff (PAS) and hematoxylin and eosin (H&E) staining were performed by the Monash Histology Platform (MHTP node) and the H&E and PAS-stained slides and were scanned using an Aperio slide scanner. Quantitative histological analysis was conducted on one complete, intact section located as close to the midline of each placenta as possible. Decidua and junctional zones were histologically identified and the area of each calculated in sections of E14.5 and E17.5 H&E-stained placentas using QuPath v0.2.3 (*Bankhead et al., 2017*). QuPath was also trained to identify glycogen-enriched and non-enriched cells using a small subsection WT-wt, HET-het, and HET-hom placental sections. Once a robust protocol had been established, machine-learning assisted image analysis provided by QuPath v0.2.3 (*Bankhead et al., 2017*) was used to quantify Glycogen cell and non-glycogen enriched cells in the junctional zone and decidua in fully intact midline sections of 10 (five male and five female) E14.5 and E17.5 placentas from WT-wt, WT-het, HET-het, and HET-hom mice. Investigators were blinded for sample genotypes throughout quantitative scoring of placental samples analysed.

## Immunohistochemistry and image analysis

Slides with 5 μm thick placental sections (described in placental histology) were baked at 60 °C for 20 min. Tissue sections were dewaxed in three changes of xylene and rehydrated in three changes of ethanol then rinsed in distilled water. Antigen retrieval was performed in DAKO PT Link in a DAKO Target Retrieval (Low pH) Solution (DAKO, Cat# S1699) at 98 °C for 30 min. Slides were then washed in DAKO EnVision Flex Wash Buffer (Cat# K8000) for 5 min. IHC was then performed on a DAKO Autostainer Plus in the following steps. Sections were washed once in EnVision Flex Wash Buffer following each subsequent step. Peroxidase Blocking Solution (DAKO, Cat# S2023) was applied for 10 min and non-specific binding was prevented with AffiniPure Fab Fragment Goat Anti-Mouse IgG for 1 hr. Mouse anti-PLAC1 (G-1) (Santa Cruz, Cat# sc-365919) primary antibody was diluted 1/100 in PBS containing 0.1% Triton X-100 (PBST) and 1% Bovine Serum Albumin (BSA, Merck) and applied to the sections for 1 hr. EnVision System-HRP Labelled Polymer Anti-Mouse (DAKO, Cat# K4001) was applied for 1 hr. Immunostaining was visualised using DAKO Liquid DAB + Substrate Chromogen System (Cat# K3468). A counterstain DAKO Automation Haematoxylin Staining Reagent was then applied for 10 min. Slides were removed from the Autostainer, transferred to a slide staining rack, and rinsed in distilled water. In a fume-hood, slides were then washed in Scott's Tap water and distilled water. Finally, slides were dehydrated in three changes of 100% Ethanol, cleared in three changes of Xylene, and mounted in DPX. Slides were scanned using a VS120 slide scanner (Olympus).

## Immunofluorescence and image analysis

Placental sections were de-waxed and processed for antigen retrieval as described above. Non-specific binding was blocked using 5% Bovine Serum Albumin (BSA, Merck) containing 10% Donkey Serum (Merck) for 1 hr at room temperature (RT). Block solution was replaced with PBST containing 1% BSA and goat anti-CD31 antibody (1/100, R&D Systems) and incubated overnight at 4 °C. Samples were washed three times in PBS before incubation with PBST containing 1% BSA and Donkey anti-Goat IgG (H+L) Cross-Adsorbed secondary antibody (Alexa Fluor 555, Thermo Fisher Scientific) for 1 hr at RT. Samples were washed three times in PBS, followed by incubation with TrueView Auto-fluorescence Quenching Kit (Vector Laboratories) for 5 min at RT according to the manufacturer's instructions. Following three washes in PBS, samples were incubated in dH2O containing 5 µg/ml DAPI (4',6-Diamidino-2-Phenylindole, Dilactate, Thermo Fisher Scientific) for 15 min at RT. Samples were washed a final three times in PBS, rinsed in dH2O, and mounted in Vectashield Vibrance Anti-fade Mounting Medium (Vector Laboratories). Slides were scanned using the VS120 Slide scanner (Olympus). Quantification of blood vessels, labyrinth area, and capillary area was achieved using machine learning-assisted image analysis using HALO (Indica Labs). Initially, HALO was trained to recognise the labyrinth area based on CD31 positive capillary staining and to exclude larger blood vessels and other tissue (e.g. the chorionic plate), which were easily distinguishable (*Figure 5B*). Then HALO was trained to calculate the area of the labyrinth occupied by CD31-positive blood vessels. Once training was complete, HALO was used to analyse all sections across 24 slides (described in placental histology) from male and female placentas of WT-wt, HET-het, and HET-hom placentas. Data was annotated in excel and statistically analysed using One-way ANOVA plus Tukey's multiple comparisons in GraphPad Prism.

## Placental RNA-sequencing and data analyses

Placental isolation is described above. RNA was extracted from 3 to 5 E17.5 placentas of both sexes for each genotype using NucleoSpin RNA Plus columns. RNA quality was assessed on an Agilent Bioanalyser and samples with RIN >7.5 used for library preparation and sequencing on the BGI Genomics platform (BGI Genomics, Hong Kong). Adaptor and low-quality sequences in raw sequencing reads were trimmed using Trimmomatic (*Bolger et al., 2014*) (v0.39) with the following parameters: LEADING:3 TRAILING:3 SLIDINGWINDOW:4:15 MINLEN:20. Clean reads were mapped to the mouse reference genome (GRCm38) using STAR (v2.7.5c) with the following settings: outFilterMismatchNoverLmax 0.03 `--alignIntronMax` 10000. Raw counts for mouse reference genes (ensembl-release-101) were calculated using STAR (v2.7.5c) with parameter '--quantMode GeneCounts' simultaneously when doing the genome mapping. Differential gene expression analysis was carried out using the R package 'limma' (*Ritchie et al., 2015*) with 'treat' function and parameter 'lfc = log(1.1).' Statistically significantly differentially expressed genes were identified using 'FDR <0.05.' Gene Ontology (GO) enrichment analysis for significantly differentially expressed genes was carried out using The Database for Annotation, Visualisation, and Integrated Discovery (DAVID) with the following settings: GO term level 3, minimum gene count 5, and FDR <0.05 (*Dennis et al., 2003*).

## Placental reduced representation bisulphite sequencing (RRBS)

5 WT-wt, 5 HET-het, and 6 HET-hom placentas from female E17.5 mice were chosen for RRBS. DNA was extracted from a quarter of each placenta with a Qiagen DNeasy Blood & Tissue Kit and eluted in AE buffer (10 mM Tris-HCl, 0.5 mM EDTA). 4–13 µg of DNA was sent to CD Genomics (NY, USA) and underwent sodium bisulphite conversion, library preparation, and Illumina PE150 sequencing at 10 Gb raw data per sample. Sequence analyses were performed on the Galaxy Australia Bioinformatics Platform (https://usegalaxy.org.au/) (*Afgan et al., 2022*). Sequence files were aligned to the GRC39 (mm39) mouse genome with BWA-meth. Per-base CpG methylation metrics were extracted with MethylDackel with the output limited to CpG sites with a minimum of five reads coverage with output presented as CpG methylation fractions. Differentially methylated CpG-rich regions (DMRs) between the groups were identified with Metilene with settings of a minimum of 10 CpGs per DMR and a minimum of 10% methylation difference between groups. Statistically significant DMRs which were below the threshold of Bonferroni adjusted p-value of q<0.05 were examined further. DMR regions were visualised and nearby genes and orthologous genome regions identified with the UCSC

Genome Browser (https://genome.ucsc.edu/ with Table Browser and LiftOver tools) or Ensemble Biomart (https://asia.ensembl.org/info/data/biomart/index.html).

## Metabolomics

E17.5 fetuses were isolated from pregnant females and decapitated on ice. ~50 ul of blood was collected from each fetus using a pipette and transferred into tubes containing 2 ul 0.5 M EDTA. collected by centrifugation and frozen at –80 C before metabolic analysis. Cardiac blood and serum samples were also collected from the mothers of each litter using the same approach. In addition, a spot of blood collected in the same way from each fetus and mother was directly applied to an Accu-Check Blood Glucose Test Strips and glucose concentration quantified using an Accu-Check Monitor. Mass spectrometry was performed on 10 ul of serum by Metabolomics Australia and included targeted profiling of 356 metabolites using the Shimadzu GCMS 8050 system. 1 ul of each sample was injected and analysed using internal standards $^{13}C_5$, $^{15}N_1$ Valine, and $^{13}C_6$ Sorbitol. 85 samples and 19 pooled blood quality controls (PBQCs) were analysed, with PBQCs used to ensure technical consistency of the instrument and sample quantifications. Samples were analysed in randomised order, with a PBQC run after every five samples. 160 metabolites were detected with 85% resulting in a coefficient of variation of <30%. Data was analysed using Metaboanalyst 5.0 to assess data quality and multivariate analyses were used to generate principal component analyses, heatmap, and volcano plots. Statistical differences between samples were assessed using an FDR <0.05 to ensure differences detected with 95% confidence after multiple comparisons correction.

## Acknowledgements

We thank A/Prof John McBain for generously supporting this work. We also thank Prof. Marnie Blewitt for critical comments on the manuscript, the Hudson Institute Animal Research Platform staff for assistance with mouse care, the Monash Histology Platform for assistance with sample preparation and slide scanning, and the Monash Micro Imaging Facility and Hudson Genomics Facility for assistance and technical advice. This project used NCRIS-enabled Metabolomics Australia infrastructure at the University of Melbourne and funded through BioPlatforms Australia. We would like to thank Dr. David De Souza and Dr. Nadeem Elahee Doomun for their advice and for running the metabolomics analyses in this study. This work was supported by grants and research funds from: National Health and Medical Research Project and Ideas Grants GNT1144966 (PSW, DKG, MvdB, DLA), GNT1144887 (PSW, DKG, DLA), and GNT2021247 (PSW, DLA), Hudson Institute of Medical Research, Victorian Government's Operational Infrastructure Support Program, Australian Government Research Training Program Scholarship support to EGJ, RO, and SP, and a philanthropic donation from Associate Professor John McBain. Metabolomics Workbench is supported by NIH U2C-DK119886 and OT2-OD030544.

## Additional information

### Funding

| Funder | Grant reference number | Author |
| --- | --- | --- |
| National Health and Medical Research Council | GNT1144966 | Patrick S Western<br>David K Gardner<br>Maarten van den Buuse<br>David L Adelson |
| National Health and Medical Research Council | GNT1144887 | Patrick S Western<br>David K Gardner<br>David L Adelson |
| National Health and Medical Research Council | GNT2021247 | Patrick S Western<br>David L Adelson |
| Australian Government | Research Training Program Scholarship | Ellen G Jarred<br>Ruby Oberin<br>Sigrid Petautschnig |

| Funder | Grant reference number | Author |
|--------|------------------------|--------|

The funders had no role in study design, data collection and interpretation, or the decision to submit the work for publication.

## Author contributions

Ruby Oberin, Sigrid Petautschnig, Thi T Truong, Conceptualization, Data curation, Formal analysis, Validation, Investigation, Visualization, Methodology, Writing – original draft, Writing – review and editing; Ellen G Jarred, Conceptualization, Data curation, Formal analysis, Validation, Investigation, Methodology, Writing – original draft, Project administration, Writing – review and editing; Zhipeng Qu, Neil A Youngson, Data curation, Formal analysis, Investigation, Methodology, Writing – original draft, Writing – review and editing; Tesha Tsai, Conceptualization, Data curation, Formal analysis, Investigation, Methodology; Gabrielle Pulsoni, Data curation, Formal analysis, Investigation, Methodology; Dilini Fernando, Data curation, Formal analysis; Heidi Bildsoe, Formal analysis, Investigation; Rheannon O Blücher, Data curation, Investigation; Maarten van den Buuse, Resources, Supervision, Funding acquisition, Writing – review and editing; David K Gardner, Resources, Formal analysis, Funding acquisition, Investigation, Methodology, Writing – review and editing; Natalie A Sims, Resources, Formal analysis, Supervision, Funding acquisition, Investigation, Writing – review and editing; David L Adelson, Resources, Data curation, Formal analysis, Supervision, Funding acquisition, Investigation, Methodology, Writing – review and editing; Patrick S Western, Conceptualization, Resources, Data curation, Formal analysis, Supervision, Funding acquisition, Validation, Investigation, Methodology, Writing – original draft, Project administration, Writing – review and editing

## Author ORCIDs

Ellen G Jarred ⓘ http://orcid.org/0000-0002-5394-9995
David K Gardner ⓘ http://orcid.org/0000-0003-3138-8274
Natalie A Sims ⓘ http://orcid.org/0000-0003-1421-8468
David L Adelson ⓘ http://orcid.org/0000-0003-2404-5636
Patrick S Western ⓘ http://orcid.org/0000-0002-7587-8227

## Ethics

All animal work was undertaken in accordance with Monash University Animal Ethics Committee (AEC) approvals issued by Monash University and Hudson Institute Animal Ethics Committees (AEC), approval numbers MMCB/2018/16 and MMCB/2020/37.

## Decision letter and Author response

Decision letter https://doi.org/10.7554/eLife.81875.sa1
Author response https://doi.org/10.7554/eLife.81875.sa2

# Additional files

## Supplementary files

• Supplementary file 1. Summary of genomic and metabolomic data collected from offspring generated from Eed wild-type (*Eed*-wt), Eed heterozygous (*Eed*-het), and Eed homozygous (*Eed*-hom) oocytes. (**A**) *Eed* Placenta differentially expressed genes (DEGs) - List of significant *Eed* female Placenta DEGs (HET-hom vs HET-het; false discovery rate, FDR <0.05; 2083 DEGs) (**B**) Differential Gene Expression Analysis Male HET-hom Placenta vs HET-het Placenta (No FDR; p<0.05) (**C**) Differential Gene Expression Analysis Female HET-hom vs HET-het Placenta (No FDR; p<0.05) (**D**) Genes Commonly Differentially Expressed in Male and Female HET-hom vs HET-het Placenta (No FDR; *P*<0.05) (**E**) *Eed* Placenta DEGs vs E14.5 single-cell RNA-seq: Comparison of *Eed* Placenta DEGs list with published single-cell RNA-seq data from E14.5 C57BL/6 placentas (**F**) *Eed* Placenta DEGs vs Imprinted genes: Comparison of *Eed* Placenta DEGs vs Mouse Imprinted genes (**G**) *Eed* Placenta DEG relative enrichment per chromosome (**H**) *Eed* Placental DMRs HET-het vs HET-hom: List of significant *Eed* female placenta differentially methylated regions (FDR/q<0.05). (**I**) *Eed* Placental DMRs WT-wt vs HET-hom: List of significant *Eed* female placenta differentially methylated regions (FDR/q<0.05). (**J**) *Eed* Placental DMRs WT-wt vs HET-het: List of significant *Eed* female placenta differentially methylated regions (FDR/q<0.05). (**K**) Metabolites detected at significantly different levels between E17.5 female HET-hom and HET-het fetal blood samples (FDR <0.05) (**L**) Metabolites detected at significantly different levels between E17.5 female HET-hom and WT-wt

fetal blood samples (FDR <0.05) (**M**) Metabolites detected at significantly different levels between E17.5 female HET-het and WT-wt fetal blood samples (FDR <0.05).

• MDAR checklist

## Data availability

All RNA sequencing and RRBS data have been deposited to the Gene Expression Omnibus (GEO) and are publicly available with accession number GSE210398. The metabolomics data are available at the NIH Common Fund's National Metabolomics Data Repository (NMDR) website, the Metabolomics Workbench, https://www.metabolomicsworkbench.org where it has been assigned Study ID ST003211. The data can be accessed directly via its Project DOI: http://doi.org/10.21228/M8TR5T. Data for Figures 6 and 8 are included in Supplementary file 1.

The following datasets were generated:

| Author(s) | Year | Dataset title | Dataset URL | Database and Identifier |
|---|---|---|---|---|
| Oberin R, Petautschnig S, Jarred EG, Qu Z, Tsai T, Youngson NA, Pulsoni G, Truong TT, Fernando D, Bildsoe H, Blucher RO, van den Buuse M, Gardner DK, Sims NA, Adelson DL, Western PS | 2024 | Fetal growth delay caused by loss of non-canonical imprinting is resolved late in pregnancy and culminates in offspring overgrowth | https://www.ncbi.nlm.nih.gov/geo/query/acc.cgi?acc=GSE210398 | NCBI Gene Expression Omnibus, GSE210398 |
| Oberin R, Petautschnig S, Jarred EJ, Qu Z, Tsai T, Youngson NA, Pulsoni G, Truong TT, Fernando D, Bildsoe H, Blucher RO, van den Buuse M, Gardner DK, Sims NA, Adelson DL, Western PS | 2024 | Fetal growth delay caused by loss of non-canonical imprinting is resolved late in pregnancy and culminates in offspring overgrowth | https://doi.org/10.21228/M8TR5T | Metabolomics Workbench, 10.21228/M8TR5T |

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
