## [Editor Report]

This important study shows that a lack of Polycomb-dependent epigenetic programming in the oocyte and early embryo influences the developmental trajectory through gestation in the mouse. The authors provide convincing evidence for a two-phase outcome of early growth restriction followed by enhancement, addressing previous inconsistencies in the field. The work establishes a link between the function of a protein in oocytes and programming of fetal growth and placental function in late gestation, though the underlying functional relationshsip between increased fetal growth and placental function will require further investigation. This manuscript will interest scientists within the fields of developmental biology and epigenetics.

---

## [Decision Letter]

**Decision letter after peer review:**

Thank you for submitting your article "Loss of EED in the oocyte causes initial fetal growth restriction followed by placental hyperplasia and offspring overgrowth" for consideration by *eLife*. Your article has been reviewed by 3 peer reviewers, and the evaluation has been overseen by a Reviewing Editor and Carlos Isales as the Senior Editor. The following individual involved in the review of your submission has agreed to reveal their identity: Marika Charalambous (Reviewer #3).

Essential revisions (for the authors):

The three reviewers recognized the merit of the work and its potential for reconciling previously conflicting results about the role of maternally inherited H3K27me3 on a growth trajectory. However, they also shared strong concerns about the interpretation of the data and in particular, the existence of a placenta-dependent catch-up phase. Additionally, it was not formally demonstrated that H3K27me3 patterns are altered in the placenta of embryos derived from EED-null oocytes, making it difficult to understand direct and indirect gene expression changes.

Please find below four points that require necessary revision. In addition, answer the individual comments of the reviewers in a detailed rebuttal letter and tone down or revise conclusions in your text accordingly.

1. Provide a more in-depth characterization of the placenta phenotype. Please first revise placental efficiency measurements as stated by Reviewer #3. Then, are there some sex-specific differences in placental alterations (considering that the placental transcriptome is differentially affected in males and females from EedKO oocytes)? Is the effect limited to the junctional zone only, and in particular to glycogen cells? The increase in the endocrine compartment may have been emphasized by the chosen methodology. Improved calculation of the different layers, within the same gestational age (E17.5) is required.

2. Whether the catch-up phase may be due to placental compensation prior to birth requires better support, especially in face of the increased gestational length and different post-coitum days of birth between mutants and WT. So far, enhanced growth is reported to happen between E17.5 and P3. The catch-up phase may equally happen prior to birth, in link with a potential placental role, or after birth, in relation to increased feeding, for example. It is really necessary to document when the catch-up phase occurs relative to birth. A measure at E19.5 was done but is only presented in Sup material: there, there is no difference between WT and mutants, which would suggest that extended gestation itself may lead to overgrowth. Please be clearer and report data in post-coitum days rather than postnatal days for WT and mutants, and add additional time points if necessary. Also, there seems to be no former evidence that placentomegaly can enhance fetal growth, and there could equally be an intrinsic fetal contribution to the phenotype.

3. Provide more clarity in the methods section, as descriptions are currently too limited to allow a proper understanding of the work carried out.

4. To claim potential epigenetic programming of placental phenotype and gene expression via Polycomb-dependent mechanisms, H3K27me3 mapping (ChIP-seq or CUT&RUN) would be required. This should be performed on placental samples at the stage when RNA-seq was done. Correlation analyses between gene expression and H3K27me3 changes (if any) are needed. It would be also interesting to compare with H3K27me3 patterns present in the oocyte (for which ChIP-seq data are publicly available).

*Reviewer #1 (Recommendations for the authors):*

In this current version, the weaknesses of the paper far outweigh the strengths. I believe further investigations are needed to improve the manuscript, in particular by validating several of the claims and conclusions. Accordingly, the authors are encouraged to consider the following (please see the justification for most of these under 'specific points'):

a) Provide a more in-depth characterization of the placenta hyperplasia phenotype to answer unequivocally the following questions:

- Is the surface area for nutrient transport increased in hom mutants vs het mutants? Suggested type of experiments – stereological measurements of the labyrinthine layer looking at fetal capillaries, maternal blood spaces, trophoblast area, and interhemal membrane thickness (a thinner membrane might indicate increased diffusion of nutrients). (Note: The need for these experiments depends on the outcome of other preliminary measurements such as volume calculations as suggested in specific points)

- Is transplacental flux increased in the hyperplastic placentas? This can be done by placental transfer assays with radiolabelled isotopes (glucose; amino acids – e.g. MeAIB; markers of diffusion such as mannitol injected to the maternal tail vein). Measuring amino-acid concentrations in fetal plasma and fetal/maternal glucose gradients would also help to provide evidence that the placenta expansion is indeed responsible for fetal catch-up growth.

- Is the junctional zone hyperplasia restricted to glycogen cells? Measuring the area occupied by spongiotrophoblasts would answer this question and add value to the paper.

- Is there evidence for sexual dimorphism related to the placenta hyperplasia phenotype? The authors did a transcriptomic analysis on male and female placentas and found very few changes in male placentas but widespread alterations in female placentas. It will be important to show that male and female placentas are hyperplastic to the same degree, so providing weights and crude histology for male vs female placentas is suggested.

- Do hyperplastic placentas show dysregulated epigenetic marking? Given that the gist of the paper is to show oocyte epigenetic programming impacting placental development, it would be important to show if there is an association with altered epigenetic marking, H3K27me3/DNA methylation (e.g. global levels, or on target genes thought to cause hyperplasia)

b) Provide alternative explanations and increase critical appraisal of data

- Why is the impact of oocyte epigenetic programming on fetal organs dismissed? It is possible that fetal organs become hyperplasic after the placenta at E17.5 and that epigenetic de-regulation also affects fetal organs? In that regard, is there evidence for disproportionate organ growth at P0 and P3?

- Could fetal and postnatal overgrowth be a secondary outcome of the increased gestational length, and therefore unrelated to placental hyperplasia and hypothetical increase in nutrient supply? It would help to show the fetal and placental weights, as well as P0 body weights, of the litters with normal gestational length vs +1 day, +2 days. If the effect on fetal catch-up growth is driven by placental hyperplasia it might be expected that offspring with normal gestational length are indistinguishable in weight from those with increased gestation length. A stratification of the weight data by gestational length and sex would clarify some of these issues.

Specific points

Data presented in Figure 4B-E shows that the junctional zone (Jz) is disproportionately bigger at E17.5 in Homozygous mutants in relation to the labyrinth (Lz) and maternal decidua, compared to control genotypes. Figure Supp Figure 3G suggests that the labyrinth may also show evidence of hyperplasia when comparing het vs hom as Lz raw area is increased but this is not the case when compared to wt (wt-het) controls. Therefore, the question of whether the phenotype is restricted to the Jz, and the glycogen cells, in particular, remains unresolved and requires more in-depth investigations by calculating volumes of the different layers, adding Jz to Lz ratios as graphs, and depending on the results then proceed with further stereological measurements within the Lz to confirm hyperplasia of this layer.

Please explain why only two groups were compared for the data that generated Figure 1B-F, why 3 groups were used in Figure 1G, and then for most of the remaining figures comparisons with 4 groups were made.

Figure 1A (bottom panel F1) – where is the evidence that WT-wt and WT-het are epigenetically different? Moreover, this study does not directly show that HET-het and HET-hom are epigenetically different, and as such a reference to those studies should be provided next to 'epigenetically different' text in panels.

Figure 1C- The variability of the data is quite considerable (4- 5 fold) but yet the error bars are rather small for those to represent SD. Could a mistake have been made and these represent SEM instead? If yes, please check carefully all other figures and figure legends. In general, it is good practice that the primary data for each figure panel, when appropriate, is provided as supplementary data.

Figure 3 (g-i) – the graph shows placental to fetal weight ratios, which does not match the text on the Y axis (i.e. body weight/placental weight). Most importantly, the graph should show fetal to placenta (F/P) ratios (and not P/F as shown)

Figure 4G – It is important to establish the extent to which the defect is limited to glycogen cells, i.e. are the number or area occupied by spongiotrophoblasts also affected (this can be done with staining for Tpbpa)? A new graph with this information should be added to this figure.

Figure 5 – I find these calculations (D-I) potentially misleading as these only contain a subset of all the potential E17.5 DEGs. I am of the opinion that this data does not merit being in the main figure and should be interpreted with caution. Instead in the main figure, it is more important to present the top DEG genes and validations by QPCR in independent samples of those and the key genes that are thought to contribute to hyperplasia. Epigenetic analysis for these selected genes, could/should be added to improve the paper in line with providing further novelty.

Line 116 – specify the blastocyst stage: all hatched blastocysts?

Line 125 – Measuring cell proliferation would confirm the findings and add value to the paper,

Line 131 – how does Figure 1D relate to Figure 1G? i.e. this is confusing – in vitro data suggests loss of blastocyst numbers but there is no difference in the number of implantation sites in vivo. please clarify in the main text.

Line 148 – please rephrase, it gives the impression that extra-embryonic tissues are lighter, but these were not measured in isolation.

Line 166 (and throughout the text) – placental efficiency is not enhanced but instead decreased. The definition of placental efficiency is 'gram of fetus produced by gram of placenta' – if Hom placentas are overgrown from E14.5 onwards and fetuses growth restricted at least until E17.5, how can this be evidence of increased placenta efficiency if fewer grams of fetus are being produced per gram of placenta?

The authors would also require a later time point between E17.5 and E19.5 if they want to validate any transient rise in placental 'efficiency' – that data is missing in the current version. In any case, it seems clear to me that placentae up to and including E17.5 are not more efficient.

Lines 189-190 – How prevalent are the spongiotrophoblast abnormal 'projections' compared to controls? Also, Figure 3J seems redundant with Figure 4A. What is the added value of 3J?

Line 198 – specify if these are midline sections (this could be in Materials and methods).

Line 207 – Where is the evidence that the overall cross-sectional area of the Labyrinth in hom-placentas is higher compared to other genotypes? At E14.5 that is not the case and at E17.5 it does not seem to be different from WTs (wt and het; as shown in Supp Figure 3G). Further analyses are thus required. This is an important point because if the Lz layer turns out not to be increased over wild-type there is no structural basis to substantiate the claim that the hyperplastic placentas are primary determinants of the catch-up growth (and also having in mind that there is no established role for glycogen stores in the placenta in determining fetal growth).

Line 208 – Should read Supp Figure 3 (E-H)?

Lines 217-220 – this is pure speculation, better suited for discussion and limitations of the study. There is no attempt in this paper to link the increased glycogen storage with glucose handling and altered metabolic profile.

Line 227 – please provide an estimate of 'total' neonatal lethality (i.e. including gestation lengths of +1 day, + 2 days, + normal length).

Lines 240-141 – I can't follow the rationale for this. Are male placentas hyperplastic to the same extent as females? One possibility to explain the lack of transcriptomic changes is that they are not? Why not perform QPCR in a number of male placentas for those genes thought to contribute to the hyperplasic phenotype (e.g. Slc38a4, Peg 3, Sfmbt2, etc) to help rule out the issues that were raised?

Lines 271-273 – as stated before, this comparison is an indirect one and only holds strictly true if comparing layers within the same gestational age at E17.5 (not against E14.5).

Lines 281-282 and Supp table 2 – please also provide the number of 'robust' imprinting, i.e. those genes that have been unequivocally shown to be imprinted, and highlight those in Supp table 2. References should also be provided in Supp Table 2. so that the reader can check for the imprinting status of the genes. For example, Slc38a1 is annotated as an imprinted gene – where is the evidence for this and is it strong enough to be credible?

Line 850 (RE: legend to Figure 1e) – please specify the stage of the embryos.

Line 859 (RE: legend to Figure 2a)- how many embryos were analysed?

Figure 6B, C and Supp Figure 5 – please add statistical indicators of variability to graphs

Supp Figure 2D – can't find an entry for this figure in the main text?

Materials and methods

The description of the methods is not detailed enough, e.g.:

Cell allocation experiments – how was the counting/i.e.segmentation of the cells done?; please provide further details on the Propidium Iodide procedure including how the staining leaves the ICM unlabelled – is this related to the timing of staining?

Placental histology and area/proportion measurements of layers – how many cross-sections were analysed per placenta? Were these midline? Did they represent different locations within the placenta?

What is meant by raw area? Would the total area be a better description? How was the 'raw' area calculated, for example in the cases where there are significant spongiotrophoblast projections into the labyrinth?

Weight measurements – please specify that these correspond to wet weights in material and methods; Use more specific terminology throughout the manuscript – consider replacing 'offspring weight' or 'body weight' with 'fetal weight' where appropriate.

Typos – should read labyrinth instead of Labryinth (Figure 4D); epigenetically instead of epigentically (Figure 1A).

*Reviewer #2 (Recommendations for the authors):*

Overall this is a thorough assessment of the fetal growth and placental characteristics of this highly novel model in which the role of Eed in the oocyte is experimentally distinguished. The experimental plan is highly innovative and clever. Considerable new information is presented. My main concern is the interpretation of the data presented. The authors strongly conclude that this is a model of late fetal growth catch-up driven by placental hyperplasia. While this may be the case, for the reasons given, there are other interpretations. For example, a different interpretation is that the observed placental changes prolong gestation allowing for longer development in utero.

Line 69 "several genes". Could the authors name the relevant genes?

Line 71 "We proposed that loss of EED in the oocyte sets up a developmental trajectory that involves initial fetal growth restriction that is resolved by placentally driven catch-up growth", "proposed" suggests that the authors have previously made this hypothesis but there is no reference. More importantly, fetal growth is intrinsically determined. This means that the fetus has the genetic potential to reach a certain size. Fetuses that do not reach their genetic potential are considered extrinsically growth restricted. This may be due to placental insufficiency or reduced maternal nutrient availability ie catch-up growth cannot be "placentally-driven". Rather, the placenta is able to support the catch-up growth of the fetus.

Line 162 "they underwent rapid catch-up growth between E17.5 and birth and were overgrown by P3" P0 is the day of birth – were the pups fed or unfed when weighed? If possible, it might be worth separating fed and unfed to ask whether there is a difference in their relative weights ie is catch-up postnatal due to feeding or prenatal due to increased placental capacity late in gestation? The numbers may not be sufficient or fed status may not be available but it is important to make this distinction given the authors' conclusion that catch-up is all about the placenta. As mentioned later in the Results section, gestation is longer which could also account for the "catch-up".

Line 165 "and enhanced placental efficiency".

Without knowing precisely when catch-up occurred and without any functional data, it is not possible to conclude that there is "enhanced placental efficiency". Please remove or rephrase.

Line 183 "To further investigate the relationship between placental function".

The authors are not experimentally determining function. Please rephrase.

Line 184 "placental weight ratio, which is indicative of placental efficiency (30)".

Although the term "placental efficiency" is used in the literature, it is not really meaningful. There are plenty of examples where the placenta is overgrown but fetal growth is still restricted – for example when the placenta is overgrowth but disorganised or where placental overgrowth is sequestering nutrients. The authors should refer only to the F:P ratios in their Results section and then discuss the possibility of increased placental efficiency underlying their observed catch-up growth. As previously stated, without weight data from either very late in gestation or from unfed pups, it is not possible to conclude the catch-up is due to the placental changes.

Line 199 "19% increase in HET-hom placental area'.

Be clear here what is meant by "placental area". Midline section of the whole placenta?

Line 217 "As placental glycogen is converted to glucose to support fetal growth, the increased glycogen cell number in HET-hom placentas is consistent with a greater glucose requirement in the late gestational fetal catch-up growth observed in the HET-hom offspring."

Although it seems intuitive that placental glycogen is used to support fetal growth, this has not been demonstrated experimentally in any study. It is possible these stores are for the dam in support of parturition or possibly to support placental hormone production. This sentence should be rephrased. In addition, the Results section should report results rather than inferring what the results might mean.

Line 223 "Consistent with this and the greater numbers of glycogen enriched cells in HET-hom placentas, gestational length was extended by 1 day".

Why "consistent with"? ie why would "supporting support the production of hormones that promote parturition" result in a longer gestation? It has been shown that an expanded junctional zone is associated with a longer gestation but not that the presumed increase in placental hormones prolongs gestation.

Line 223 "gestational length was extended by 1 day".

Could the weight gain reported at P0 simply reflect the fact that gestation is longer in the HET-hom model?

Line 229 and elsewhere "fetal catch-up growth".

The authors need to be cautious in concluding that there is fetal catch-up growth as this has not been demonstrated.

Line 235 "we analysed male and female placental tissue from HET-hom, HET-het and WT-wt offspring" and "Surprisingly".

The authors should be commended for recognising sexual dimorphism in the placenta. Were any differences in placental regions or fetal/placental weights detected between males and females at any stage? The absence of substantial differences in gene expression would only be surprising if the authors show a similar placental HET-hom phenotype for males and females

Line 271 "Of the Eed junctional zone DEGs the majority (86.1%) were upregulated, whereas a large proportion of Eed labyrinth DEGs and Eed decidua DEGs were downregulated"

As the placental phenotype involves an expansion of the junctional zone, it is to be expected that Jz-expressed genes will be expressed at overall higher levels and Lz at lower levels simply as a consequence of the relative changes in the number of cells expressing these genes ie there is no "upregulation" or "down-regulation". Please rephrase. The authors do acknowledge that the gene changes are consistent with the relative changes in the proportion of these regions in their last sentence but it is still important to avoid active terms such as upregulation/downregulation.

Line 292 "nine were commonly dysregulated.

Do the authors mean that the direction of aberrant expression was the same? i.e. expressed at higher levels in both scenarios?

Line 300 "dysregulation detected in Eed HET-hom placenta.

Rephrase as "some of the gene expression differences detected in female Eed HET-hom placenta.

Line 308 "similar placental phenotypes".

Loss of expression of Plac1 results in Jz overgrowth. Both overexpression and loss of expression of Ascl2 result in a decreased Jz. The term "similar phenotypes" should be removed here and instead simply state that a number of imprinted genes have been genetically demonstrated to regulate Jz development.

Line 322 "and inter-related fetal and placental growth trajectories

Inter-related – what is implied by this term?

Line 332 "indicating a role for the placenta in responding to fetal growth restriction to drive offspring catch-up growth prior to birth".

This cannot be concluded from the data presented. The authors have not shown that the placenta is responding. Critically, the authors have not definitively shown catch-up prior to birth.

Line 333 "placental efficiency increased late in gestation.

F:P ratio is not a true indicator of placental efficiency. If the authors were to show fetal growth catch-up in utero, the change in F;P ratio would be "consistent with" increased placental efficiency.

Line 338 "growth restriction can be corrected by placental expansion at later stages".

It may be true that fetal growth restriction as a consequence of placental insufficiency can be corrected by placental expansion but there are plenty of examples where the opposite is true ie overgrowth of the placenta alongside fetal growth restriction.

Line 373 "this was resolved by birth".

Again, not quite demonstrated by the data presented particularly given the increased length of gestation.

Line 394 "the maternal side of the placenta also expands to accommodate greater placental function".

Again, not experimentally demonstrated. It may be the case that changes in the trophoblast lineages/placental hormone production drive increased decidualisation – not quite the same thing.

Line 400 "sustained high levels of glycogen stores in the E17.5 placentas of HET-hom offspring increase glycogen release to the embryo and drive rapid growth catch-up growth."

Firstly, the authors have not measured placental glycogen stores biochemically. More importantly, the term "drive" is over-interpreting the data presented.

Line 413 "It may be that the increased number of glycogen cells remaining in the late-gestation HET-hom placenta delays late gestational hormonal release from the placenta and prolongs pregnancy. Consistent with this, extended gestational length occurred in about two-thirds of the pregnancies from Eed-hom oocytes."

Here the authors are retrofitting their data. It may be the case that prolonged gestation might be explained by delays in hormone release but how the increased number of glycogen cells would lead to such a result is not clear at all.

Line 416 "with no apparent birthing difficulties for the mothers".

Did the authors observe or video the births? If not, they cannot conclude there were no birth difficulties. This would be possible if, for example, all the pups were observed to be viable immediately after the birth and subsequently died. Is there any data on whether the pups that died were fed or unfed?

Line 471 "fetal catch-up growth".

Again, not proven.

Discussion generally

The reduced number of males is of interest. Nothing in the discussion?

Line 534 "Junctional zone, labyrinth and decidual area".

Area or mid-line area? Please clarify in Materials and methods.

Line 516 Were males removed during the day – specify in Materials and methods.

*Reviewer #3 (Recommendations for the authors):*

Methods – more clarity is requested in this section, descriptions are often very brief.

1) Placental CSA and cell proportions – how were the sections chosen or was the placental half exhaustively sectioned and then cell counts performed? The proportions of cells in the lateral part of the placenta are different from those at the midline – authors should describe their method in more detail including the location of sections chosen. How was GlyT cell number estimated and what are the units?

2) Presentation of the placental proportion data – there is clearly an expansion of all of the placental zones, but a proportionate increase in the JZ. In terms of the interpretation of the data, this is important. Because the LZ is increased in size then there is an increased exchange volume between mother and offspring. Can the JZ expansion be accounted for by the increase in GlyT cells alone?

3) How are cells in placental zones deduced from sc-RNAseq data – this section needs more detail.

4) RNAseq – how was the list of imprinted genes derived (there seems like a lot)? Is there functional enrichment for imprinted genes in their DEG list?

5) How was cell viability performed in the live tracking experiment (Figure 1D)?

[Editors' note: further revisions were suggested prior to acceptance, as described below.]

Thank you for resubmitting your work entitled "Fetal growth delay caused by loss of non-canonical imprinting is resolved late in pregnancy and culminates in offspring overgrowth" for further consideration by *eLife*. Your revised article has been evaluated by Adèle Marston (Senior Editor) and a Reviewing Editor.

The manuscript has been improved but there are some remaining issues raised by reviewer 2 that need to be addressed, as outlined below:

In particular, the point about normalization of fetal growth in mutants being a consequence of longer gestation period or the proposed intra-uterine catch-up growth has not been adequately addressed. Evidence should be provided that timing of birth is identical between genotypes to make this claim. Without this, conclusions about fetal catch-up should be toned down, removed from abstract and the two hypotheses (gestational length/catch-up) would need to considered alongside in results & discussion, highlighting that future studies are needed to distinguish between the two.

Please also re-write the Results section for clarity, following the suggestions of Reviewer 3 below, and move the interpretations of results to the Discussion section.

*Reviewer #3 (Recommendations for the authors):*

Main points about the claims of the paper:

1) Offspring of mouse oocytes lacking EED are initially developmentally delayed and growth restricted.

Well founded and supported by the data, though this was previously established.

2) Placentas from EED-deleted oocytes are dysmorphic with increased junctional zone and glycogen cell volume.

Histologically well demonstrated, but the molecular evaluation is incomplete and muddled. There are considerable differences in gene expression between EED-mutant and control placentas. This would be expected since the cell proportions are very different. The transcriptional data does not provide much additional explanatory power, at least as currently presented.

3) Offspring undergo accelerated development and growth in late gestation.

Evidence for this claim is incomplete. I remain unconvinced that the normalisation in size between Het-hom and control offspring on the day of birth does not reflect a difference in the length of gestation. All embryo weights up to e18.5 show growth restriction in the mutants (Figure 2B-F). It is only in post-natal animals that normalisation is observed.

As far as I can tell from the methods, the estimation of gestation length is based on the observation of a copulation plug on the morning following a mating. It is not clear if animals were separated in between mating days and what time they were set up. In addition, the time of birth was not recorded. Therefore, gestational length could reflect an interval of up to 24 hours. This is unlikely since mice usually mate at the beginning of the dark cycle and give birth at the end of it. However, without monitoring the time of birth there could be a significant delay in parturition in the mutant pregnancies that could still account for the small amount of 'growth recovery' observed in this study. Without a more thorough study of birth timing, I don't feel that the authors can hang the major conclusions of their paper on the single finding that the P0 weights are similar between genotypes.

---

## [Author Response]

Essential revisions (for the authors):The three reviewers recognized the merit of the work and its potential for reconciling previously conflicting results about the role of maternally inherited H3K27me3 on a growth trajectory. However, they also shared strong concerns about the interpretation of the data and in particular, the existence of a placenta-dependent catch-up phase. Additionally, it was not formally demonstrated that H3K27me3 patterns are altered in the placenta of embryos derived from EED-null oocytes, making it difficult to understand direct and indirect gene expression changes.Please find below four points that require necessary revision. In addition, answer the individual comments of the reviewers in a detailed rebuttal letter and tone down or revise conclusions in your text accordingly.1. Provide a more in-depth characterization of the placenta phenotype. Please first revise placental efficiency measurements as stated by Reviewer #3. Then, are there some sex-specific differences in placental alterations (considering that the placental transcriptome is differentially affected in males and females from EedKO oocytes)? Is the effect limited to the junctional zone only, and in particular to glycogen cells? The increase in the endocrine compartment may have been emphasized by the chosen methodology. Improved calculation of the different layers, within the same gestational age (E17.5) is required.2. Whether the catch-up phase may be due to placental compensation prior to birth requires better support, especially in face of the increased gestational length and different post-coitum days of birth between mutants and WT. So far, enhanced growth is reported to happen between E17.5 and P3. The catch-up phase may equally happen prior to birth, in link with a potential placental role, or after birth, in relation to increased feeding, for example. It is really necessary to document when the catch-up phase occurs relative to birth. A measure at E19.5 was done but is only presented in Sup material: there, there is no difference between WT and mutants, which would suggest that extended gestation itself may lead to overgrowth. Please be clearer and report data in post-coitum days rather than postnatal days for WT and mutants, and add additional time points if necessary. Also, there seems to be no former evidence that placentomegaly can enhance fetal growth, and there could equally be an intrinsic fetal contribution to the phenotype.3. Provide more clarity in the methods section, as descriptions are currently too limited to allow a proper understanding of the work carried out.4. To claim potential epigenetic programming of placental phenotype and gene expression via Polycomb-dependent mechanisms, H3K27me3 mapping (ChIP-seq or CUT&RUN) would be required. This should be performed on placental samples at the stage when RNA-seq was done. Correlation analyses between gene expression and H3K27me3 changes (if any) are needed. It would be also interesting to compare with H3K27me3 patterns present in the oocyte (for which ChIP-seq data are publicly available).

In response to the Reviewer’s comments, we have significantly rewritten our manuscript and have added substantially more data to the study. These data include new analyses of glucose, amino acid and metabolite levels in fetal and maternal blood samples, more highly resolved fetal growth analyses, and a more detailed study of the hyperplastic placenta including IF analyses of labyrinth and capillary areas and ratios. We have also added analyses of placental DNA methylation state in offspring from oocytes lacking EED, which reveal a range of DNA methylation changes at imprinted and non-imprinted genes in *HET-hom* offspring compared to *HET-het* or *WT-wt* controls. We have also re-titled the manuscript to better reflect the findings of our revised study.

Intergenerational epigenetic inheritance in mammals remains poorly understood, particularly with respect to the contributions of histone modifications. To understand the consequences of disrupting these mechanisms, there remains a significant need for more information on the phenotypic and physiological consequences of perturbed epigenetic inheritance in pre- and post-implantation embryos and postnatal life. Our study extends understanding of intergenerational epigenetic inheritance by providing novel insights into the phenotypic and physiological consequences of failed EED-dependent programming in oocytes. This is particularly important considering that the developmental trajectory experienced by the embryo and fetus can have life-long impacts on health.

Our study was first submitted to an alternative journal in January 2022, but we were unsatisfied with the Reviewer’s comments and elected to transfer our manuscript to the open process *eLife* provides. During this time Matoba *et al.,* published a paper which demonstrated that deleting *Slc38a4* or *C2MC,* a micro-RNA cluster embedded within *Sfmbt2,* was able to rescue placental hyperplasia in offspring generated from oocytes lacking EED (Matoba *et al.*, 2022 *Genes and Development*). This study also revealed that deletion of both *Eed* and *Xist* in oocytes corrected male-biased lethality in *HET-hom* offspring, but only partially normalised litter size and offspring weight. While the outcomes reported by Matoba *et al.,* are highly informative, the rescue provided by maternal *Xist* deletion was incomplete and we hypothesise that other mechanisms must contribute to fetal growth and developmental outcomes in offspring from *Eed*-null oocytes*.*

In our study we demonstrate that surviving *HET-hom* offspring were able to attain normal weights by birth without deletion of *Xist* in the oocyte or correction of H3K27me3-dependent imprinting. Remarkably, the fetal growth delay in *HET-hom* fetuses was resolved during pregnancy despite reduced placental efficiency, low fetal blood glucose levels and unaltered amino acid and metabolite levels. Together, our observations indicate that *HET-hom* offspring have an innate ability to ameliorate fetal growth delay without increased placental function and within pregnancies of normal gestational length. Moreover, our comparison of *HET-hom* and SCNT derived growth and placental data suggest that similar fetal growth recovery occurs in both of these models without a requirement to correct maternally inherited impacts mediated by *Xist* or H3K27me3-dependent imprinting.

While the reason for this growth and developmental recovery during pregnancy remains unclear, our model demonstrates that these outcomes were initiated by loss of EED in the growing oocyte or during preimplantation development, or both. Given the capacity for fetal growth restriction to be resolved in the absence of enhanced placental function, we speculate that an unknown mechanism may sense growth restricted pregnancies to support adaptation(s) that favour the delivery of fully grown pups. In addition, we show that 60% of pregnancies are extended in length and that this increased gestational time contributes to offspring overgrowth. While it is clear that loss of H3K27me3 at *Slc38a4, C2MC* or *Xist* significantly contribute to the placental phenotype, loss of male fetuses and fetal growth delay, our study demonstrates that fetal growth and development are recovered in *HET-hom* offspring despite loss of H3K27me3 imprinting.

[Editors’ note: what follows is the authors’ response to the second round of review.]

The manuscript has been improved but there are some remaining issues raised by reviewer 2 that need to be addressed, as outlined below:In particular, the point about normalization of fetal growth in mutants being a consequence of longer gestation period or the proposed intra-uterine catch-up growth has not been adequately addressed. Evidence should be provided that timing of birth is identical between genotypes to make this claim. Without this, conclusions about fetal catch-up should be toned down, removed from abstract and the two hypotheses (gestational length/catch-up) would need to considered alongside in results & discussion, highlighting that future studies are needed to distinguish between the two.Please also re-write the Results section for clarity, following the suggestions of Reviewer 3 below, and move the interpretations of results to the Discussion section.

We have added a new analysis to demonstrate that the weight deficit in HET-hom offspring is resolved between E14.5 and E19.5 (excluding offspring with extended gestational length). Please refer to our detailed response to Reviewer 3, comment 3. We have also responded to Reviewer 3’s request to edit the Results section.

Given that the new analysis we have provided clearly demonstrates that the weight deficit is resolved in HEThom offspring between E14.5 and E19.5, we have opted not to remove our observation that offspring fetal growth is normalised within a normal gestational length from the abstract. We are confident that our conclusions are correct and our interpretation of these data is more accurately reflected in the current wording of the manuscript.

Lines 350-355 (now 355-359) describe a comparison we performed as part of our experimental work. We extracted SCNT offspring and placental weight data from the supplementary material of Matoba et al., 2018 (26) and compared these data with our own Wt-wt and Het-hom offspring and placental weight data for the oocyte *Eed*-null model described in this study. This revealed marked similarities between these models. As this is part of our analysis, we prefer to leave this in the Results section, but have now explained this experiment in more detail (please refer to our response to Reviewer 3, Minor issue 2). Lines 355-359 now read: “Given that both *Eed* HET-hom offspring and SCNT offspring lack H3K27me3 imprinting, we compared E14.5 and E19.5 fetal and placental weight data collected in this study with SCNT offspring data extracted from Matoba *et al.,* 2018 (26). This revealed very similar fetal and placental growth trajectories in the *Eed* HET-hom and SCNT offspring models (Figure 7).”

While lines 415-417 (now lines 417-421) could be moved to discussion, they are included in results as an example of a non-imprinted gene that may be functionally important in the placenta. We included the sentence “An interesting example is *NTN1* (Netrin1), which is reduced in the placentas of women with fetal growth restriction and potentially influences placental size by increasing the viability of placental microvascular endothelial cells (48, 49)” to explain why NTN DNA methylation status was highlighted in comparison to its placental expression, including reference to these data in Supplementary Table 8. We have edited this sentence and prefer to leave this as now described, though we will move the sentence to discussion if required.

To capture the context here, the text referred to on lines 414-421 now reads:

“The latter included five DMRs within or nearby eight non imprinted genes (*Ntn1*, *Unc45b*, *Fam83h*, *Tiam1*, *Pcdhga1*, *Pcdhga5*, *Pcdhgb5*, and *Pcdhgb7*) that were differentially expressed between HET-het and HET-hom placentas (Figure 6I; Supplementary File 1G). Of these, *NTN1* (Netrin1) is of particular interest as it is reduced in the placentas of women with fetal growth restriction and potentially influences placental size by increasing the viability of placental microvascular endothelial cells (48, 49). While *Ntn1* is not imprinted, it had reduced DNA methylation, consistent with its increased transcription in HET-hom compared to HET-het placentas (Supplementary File 1H).”

Reviewer #3 (Recommendations for the authors):Main points about the claims of the paper:1) Offspring of mouse oocytes lacking EED are initially developmentally delayed and growth restricted.Well founded and supported by the data, though this was previously established.

We appreciate Reviewers comments. While previous studies have noted developmental delay, there has been no attempt to study this in detail or to explain how overgrowth could be observed in one study (our work in Prokopuk et al., 2018) while developmental delay was observed in another (Inoue et al., 2018). In this study we have taken great care to characterise the fetal growth in the context of the placental phenotype and have provided significantly more detail than previous studies. This includes determining when growth delay is initiated in preimplantation embryos and its eventual resolution during the later stages of pregnancy.

2) Placentas from EED-deleted oocytes are dysmorphic with increased junctional zone and glycogen cell volume.Histologically well demonstrated, but the molecular evaluation is incomplete and muddled. There are considerable differences in gene expression between EED-mutant and control placentas. This would be expected since the cell proportions are very different. The transcriptional data does not provide much additional explanatory power, at least as currently presented.

We assume that by molecular evaluation of the placenta, reviewer 2 refers to the RNA sequencing and DNA methylation analyses. In our revised manuscript, as well as the DNA methylation analysis in placenta, we added analyses of metabolic state and circulating glucose levels in fetal and maternal blood samples and a detailed analysis of placental vasculature content using IF. These additional data substantially added to our initial analyses and have a bearing on the potential worth of additional molecular analyses in HET-hom placentas.

In HET-hom offspring we found substantially larger placentas (hyperplasia) and smaller fetuses, which are indicative of lower placental function assessed by fetal / placenta weight ratio. However, we found no difference in metabolic state in the serum of fetal offspring or in their mothers, strongly suggesting that the placenta does not have a major impact on metabolic state in the blood of the fetus or the mother. Glucose levels were also similar in fetuses and mothers at E17.5, although the average level was lower at E18.5, an observation most likely explained by the delay in parturition in around 60% of litters. Together these data indicate that despite the clear placental hyperplasia, metabolic state was not altered in HET-hom offspring blood samples and it is therefore difficult to attribute changes in late mid-gestational fetal growth to altered placental function at the metabolic level.

We appreciate that the global transcriptional analyses of the placenta lack spatial power and changes in cell proportions in the placenta will also contribute to overall changes in transcription. The way to resolve this would be by single cell or spatial RNAseq. However, if we performed single cell or spatial analyses, any cell specific transcriptional changes identified must still be interpreted in the light of the metabolic analyses. Given that we did not observe changes (either up or down) in the metabolic state of *HET-hom* offspring or their mothers, it is difficult to see how more detailed transcriptional analyses of the placental data could then be used to explain how the developmental delay and weight deficit that was observed at E14.5 in HET-hom offspring was resolved by birth (further discussed in our response to Reviewer 3, comment 3).

Single cell and spatial analyses are very expensive and time consuming. Moreover, given our findings, such an approach would have to include detailed analyses of fetal tissues as well as the placenta as, given the placental data, the fetal growth recovery we observed is likely to have at least some origin in the fetus. This would take substantial time and resources and would be unlikely to increase our ability to explain the central outcomes in this model. We therefore consider this to be outside the scope of the current study.

3) Offspring undergo accelerated development and growth in late gestation.Evidence for this claim is incomplete. I remain unconvinced that the normalisation in size between Het-hom and control offspring on the day of birth does not reflect a difference in the length of gestation. All embryo weights up to e18.5 show growth restriction in the mutants (Figure 2B-F). It is only in post-natal animals that normalisation is observed.

Our data do not support the conclusion that growth normalisation was only observed in post-natal animals. The data we provided must be taken in context of the whole developmental and growth curve of these offspring, from early embryonic stages until birth. We observed very marked developmental delay and weight deficit of HET-hom offspring at E9.5, E12.5 and at E14.5 (Figure 2A-D). At E17.5 HET-hom fetuses were still underweight compared to controls (Figure 2E). However, the difference at E17.5 was reduced compared to E14.5 and, while there remained a small difference in fetal weight between HET-hom offspring and controls at E18.5 (Figure 2F), the difference was modest. Moreover, while HET-hom embryos were very obviously morphologically delayed at E9.5 (Figure 2A), E12.5 (Figure 2B) and E14.5 (Figure 2 supplement 1B), with the exception of weight, it was no longer possible to morphologically distinguish HEThom offspring at the gross morphological level at E18.5. Finally, the weight of HET-hom offspring born on E19.5 was the same as controls (Figure 2j-K).

We have now illustrated these data by providing ratios of HET-hom offspring weight vs WT-wt, WT-het and HET-het control weight at E12.5, E14.5, E17.5, E18.5 and E19.5. Importantly, E19.5 data included only litters born on E19.5. (New Figure 2L). This clearly demonstrates that compared to all control genotypes, HET-hom weight was normalised between E14.5 and E19.5.

To reflect this analysis, we have added the following text and edits to lines 184-194 of the Results section: “However, ratios of HET-hom fetal weight over WT-wt, WT-het or HET-het fetal weight at E12.5, E14.5, E17.5, E18.5 and E19.5 (excluding litters of extended gestational length) revealed that the weight deficit in HET-hom was resolved between E14.5 and birth on E19.5 (Figure 2L). Moreover, with the exception of the modest weight deficit, E18.5 HET-hom offspring were indistinguishable from controls at the gross morphological level. Together, these data demonstrate that the gross morphological delay and the fetal weight deficit in HET-hom offspring was resolved by E19.5 and that this occurred independently of litter size. In addition, by P3 HET-hom offspring were heavier than controls. As the overgrown HET-hom offspring in the P3 weight cohort (Figure 2H) included litters born at E19.5 E20.5 and E21.5, both post-natal offspring growth and extended gestation presumably contributed to the overgrowth phenotype.”

Our observations clearly show that the developmental delay and weight deficit are resolved in HET-hom offspring between E14.5 and birth on E19.5. Our data do not support the conclusion that resolution of developmental delay and weight deficit occurred as a result of the extended gestational length in HET-hom offspring. However, it is clear that extended gestational length contributed to the overgrowth phenotype we observed in this study, and that was observed in our previous study (Prokopuk et al., 2018).

Previous studies have not examined the apparently discordant observations that offspring from *Eed* null oocytes are delayed during fetal development (Inoue et al., 2018, Matoba et al., 2022), but are overgrown post-natally (Prokopuk et al., 2018). Here we now demonstrate that these observations are concordant and provide clear evidence that developmental delay and the deficit in fetal weights were resolved during gestation and that offspring are ultimately overgrown partly as a result of extended gestation.

Moreover, by discovering that deletion of *Eed* in oocytes results in extended gestational length in some litters we reveal a fascinating aspect of this model that had previously remained unknown.

As far as I can tell from the methods, the estimation of gestation length is based on the observation of a copulation plug on the morning following a mating. It is not clear if animals were separated in between mating days and what time they were set up. In addition, the time of birth was not recorded. Therefore, gestational length could reflect an interval of up to 24 hours. This is unlikely since mice usually mate at the beginning of the dark cycle and give birth at the end of it. However, without monitoring the time of birth there could be a significant delay in parturition in the mutant pregnancies that could still account for the small amount of 'growth recovery' observed in this study. Without a more thorough study of birth timing, I don't feel that the authors can hang the major conclusions of their paper on the single finding that the P0 weights are similar between genotypes.

We have performed studies using standard time-mating protocol for more than 20 years and have a great deal of experience in staging in embryos and fetuses. We used the same protocol to generate the litters examined in this study and to test whether gestational length was affected in the *Eed* model. Our protocol included checking for copulation plugs early on the morning following mating and immediately separating all females from males when a plug was discovered. To ensure we did not miss the day of birth, all females were visually monitored for litters without disturbance twice a day – first thing in the morning and last thing in the afternoon.

We apologise for omitting the important detail that females were separated when a plug was detected in timed mates, but have now added this in Materials and methods.

The methods now read (Lines 651-656) “Mice were time mated for two-four nights, with females plug checked daily for copulation plugs. Positive plugs were noted as day E0.5 and all females for which a plug was discovered were immediately separated from the male. Gestational length was measured in days post copulation by recording the morning of a copulation plug was detected E0.5 and visually monitoring females twice daily (morning and afternoon) for births from late gestation (E18.5) until pups were delivered (E19.5E21.5, depending on oocyte genotype).”

The data produced for gestational length are very clear. Extended gestation was only detected in litters from *Eed-*null oocytes (i.e. HET-hom offspring). In these animals 40% of litters were born on E19.5, 30% were born on E20.5 and 30% were born on E21.5. All litters (100%) from wild type or heterozygous oocytes were born on E19.5. Moreover, given that we monitored births in the morning and evening there could be a maximum of a half day difference for the 40% of litters born on E19.5 and for most the difference is less than that.

The data we have presented for E12.5-E19.5 offspring weights, including the new analysis we provide in Figure 2L, clearly demonstrate that the weight deficit of HET-hom offspring was resolved between E14.5 and E19.5. To conclude that the HET-hom weight deficit was resolved in the very last few hours of the pregnancy is not consistent with our data, especially given that this period also involved delivery.

To our knowledge, no other study has monitored gestational length in offspring from *Eed*-null oocytes or in studies of H3K27me3 imprinting in SCNT offspring. Our study is the first and in doing so provides important insight into the physiological responses that may help Het-hom offspring survive.

For all genotypes (experimental and control) there is some variation in plugging time, fertilisation time and birth time in mice and all of these factors influence exact gestational length. Measuring exact fertilisation time would require IVF followed by embryos transfer, rather than natural mates. However, IVF followed by embryo transfers may also affect gestational length as the procedure, timing of implantations and number of implantations are likely to vary between recipient females. In our opinion using IVF would be inferior to using natural mating to produce offspring in this model.

The most effective way to properly observe the exact hour of birth would be to place cameras in cages. This is impractical and would be hampered by cage contents such as tissues, wood shavings, nesting material etc, which enrich the female’s birthing environment. Such an approach would require repetition of all the term deliveries performed in this study. Moreover, it would be very expensive and time consuming, questionable from practical and ethical viewpoints and is unlikely to enhance the interpretation of the data that we have already provided.